# Resistance to lethal ectromelia virus infection requires Type I interferon receptor in natural killer cells and monocytes but not in adaptive immune or parenchymal cells

**Carolina R. Melo-Silva**[ID], **Pedro Alves-Peixoto**[ID][¤a], **Natasha Heath**[ID][¤b], **Lingjuan Tang**[ID], **Brian Montoya**[ID], **Cory J. Knudson**[ID], **Colby Stotesbury**[ID][¤b], **Maria Ferez**[¤c], **Eric Wong**[ID][¤b], **Luis J. Sigal**[ID]*

Department of Microbiology and Immunology, Thomas Jefferson University, Philadelphia, Pennsylvania, United States of America

¤a Current address: Life and Health Sciences Research Institute, School of Medicine, University of Minho, Braga, Portugal
¤b Current address: GlaxoSmithKline, Collegeville, Pennsylvania, United States of America
¤c Current address: Spark Therapeutics, Philadelphia, Pennsylvania, United States of America
* Luis.Sigal@jefferson.edu

**Data Availability Statement:** All relevant data are within the manuscript and its Supporting Information files.

## Abstract

Type I interferons (IFN-I) are antiviral cytokines that signal through the ubiquitous IFN-I receptor (IFNAR). Following footpad infection with ectromelia virus (ECTV), a mouse-specific pathogen, C57BL/6 (B6) mice survive without disease, while B6 mice broadly deficient in IFNAR succumb rapidly. We now show that for survival to ECTV, only hematopoietic cells require IFNAR expression. Survival to ECTV specifically requires IFNAR in both natural killer (NK) cells and monocytes. However, intrinsic IFNAR signaling is not essential for adaptive immune cell responses or to directly protect non-hematopoietic cells such as hepatocytes, which are principal ECTV targets. Mechanistically, IFNAR-deficient NK cells have reduced cytolytic function, while lack of IFNAR in monocytes dampens IFN-I production and hastens virus dissemination. Thus, during a pathogenic viral infection, IFN-I coordinates innate immunity by stimulating monocytes in a positive feedback loop and by inducing NK cell cytolytic function.

## Author summary

Type I interferon (IFN-I) is required for resistance to many viral infections and the IFN-I receptor IFNAR is ubiquitously expressed in most cells. Melo-Silva et al. show that resistance to mousepox, a highly lethal viral disease of mice, requires IFNAR in Natural Killer cells to promote optimal maturation and cytotoxicity, and in inflammatory monocytes for a positive feedback loop necessary to induce optimal IFN-I expression. However, intrinsic IFNAR is dispensable in adaptive lymphocytes or parenchymal cells indicating that for at least some viral infections, the critical anti-viral effects of IFN-I are restricted to cells of the innate immune system and are superfluous in other cell types.

**Funding:** This work was supported by grants to: 1) LJS, from the National Institute of Allergy and Infectious Diseases (https://www.niaid.nih.gov) R01AI110457, R01AI065544, and T32AI34646, which supported the salaries of BM and CJK; from the National Institute on Aging (https://www.nia.nih.gov) R01AG048602; and from the American Association of Immunologists ( https://www.aai.org), Careers in Immunology Fellowship, which partly supported the salary of CRM). 2) EW, the National Institute of Allergy and Infectious Diseases (https://www.niaid.nih.gov) F32AI134646), 3) Sidney Kimmel Cancer Center at Thomas Jefferson University, the National Cancer Institute (https://www.cancer.gov) P30CA056036, for the use of Flow Cytometry and Laboratory Animal Facilities. The funders had no role in study design, data collection and analysis, decision to publish, or preparation of the manuscript.

**Competing interests:** No. The authors have declared that no competing interests exist.

## Introduction

Type I interferons (IFN-I) are a family of highly conserved antiviral cytokines. IFN-I bind to the IFN-I receptor IFNAR, a heterodimer composed of IFNAR1 and IFNAR2. Binding of IFN-I to IFNAR induces a signaling cascade that culminates in the upregulation of interferon-stimulated genes (ISG) [1]. ISGs induce an antiviral state in cells that curtail viral replication [2]. IFN-I is also thought to activate innate and adaptive immune responses through direct and indirect IFNAR signaling. Consequently, resistance to many viral pathogens requires IFNAR as determined by infection of mice deficient in IFNAR ($Ifnar1^{-/-}$) [3–5] and increased adverse reactions to attenuated viral vaccines observed in humans with inherited IFNAR deficiency [6].

The discovery of many mechanisms for IFN-I induction was made in fibroblasts [7]. Also, it is well established that virtually any cell type can produce IFN-I *in vitro*. However, it is now apparent that *in vivo*, the cells that mostly produce IFN-I are myeloid lineage, such as conventional dendritic cells (cDC), plasmacytoid dendritic cells (pDC), or Ly6C$^{high}$ inflammatory monocytes (iMOs) [8]. Similarly, most cell types express IFNAR, and when exposed to IFN-I *in vitro*, upregulate ISGs and become more resistant to viral infection [2]. Therefore, the general thought is that the stimulation of IFNAR by IFN-I is protective as it can induce ISGs and the antiviral state in virtually any cell, and because it activates multiple innate and adaptive immune cell types [9]. However, whether this is true *in vivo* has not been thoroughly studied.

Ectromelia virus (ECTV) is the etiologic agent of mousepox, which causes fulminant hepatitis in mice and resembles human smallpox. C57BL/6 (B6) mice are naturally resistant to lethal mousepox, and this resistance requires IFN-I, which in the draining lymph node (dLN) is mostly produced by infected iMOs recruited by dendritic cells (DCs) [10,11]. Defects in the IFN-I response *in vivo* such as the absence of IFNAR [12,13], the DNA sensor Cyclic GMP-AMP synthase (cGAS) [14,15], the signaling adapter Stimulator of Interferon Genes (STING) or the transcription factor Interferon regulatory factor 7 (IRF7) [11], results in unchecked ECTV replication, impairment of the adaptive immune response, and death.

Given that IFN-I signaling is crucial for mousepox resistance, it is an excellent model to uncover which cells need to receive IFN-I signaling through IFNAR to resist a lethal viral infection, and to determine whether hematopoietic and non-hematopoietic cells require intrinsic IFNAR. This question is especially intriguing in the context of hepatocytes, which are non-hematopoietic and a key target of ECTV pathogenesis. Furthermore, considering that resistance to mousepox needs a robust cytotoxic response by both innate Natural Killer (NK) and adaptive T cells, it is important to investigate whether resistance to viral disease requires intrinsic IFNAR in these cell types. Also, given the critical role of iMOs in IFN-I production, it is of interest to define the role of IFNAR in these cells.

## Results

### IFNAR in hematopoietic cells is necessary and sufficient for resistance to lethal mousepox

To test whether hematopoietic or non-hematopoietic cells require IFNAR signaling for resistance to mousepox, we lethally irradiated B6 and $Ifnar1^{-/-}$ mice and reconstituted them with either B6 or $Ifnar1^{-/-}$ bone marrow cells to generate B6→ B6, $Ifnar1^{-/-}$→ B6, B6→ $Ifnar1^{-/-}$ and $Ifnar1^{-/-}$→ $Ifnar1^{-/-}$ bone marrow chimeras (BMC). Following ECTV infection in the footpad, all $Ifnar1^{-/-}$→ B6 and $Ifnar1^{-/-}$→ $Ifnar1^{-/-}$ mice succumbed rapidly to the infection, whereas most B6→ $Ifnar1^{-/-}$ and B6→ B6 mice survived (**Fig 1A**). IFNAR deficiency restricted to radio-resistant cells resulted in only mild susceptibility to mousepox, given that ~70% of B6→

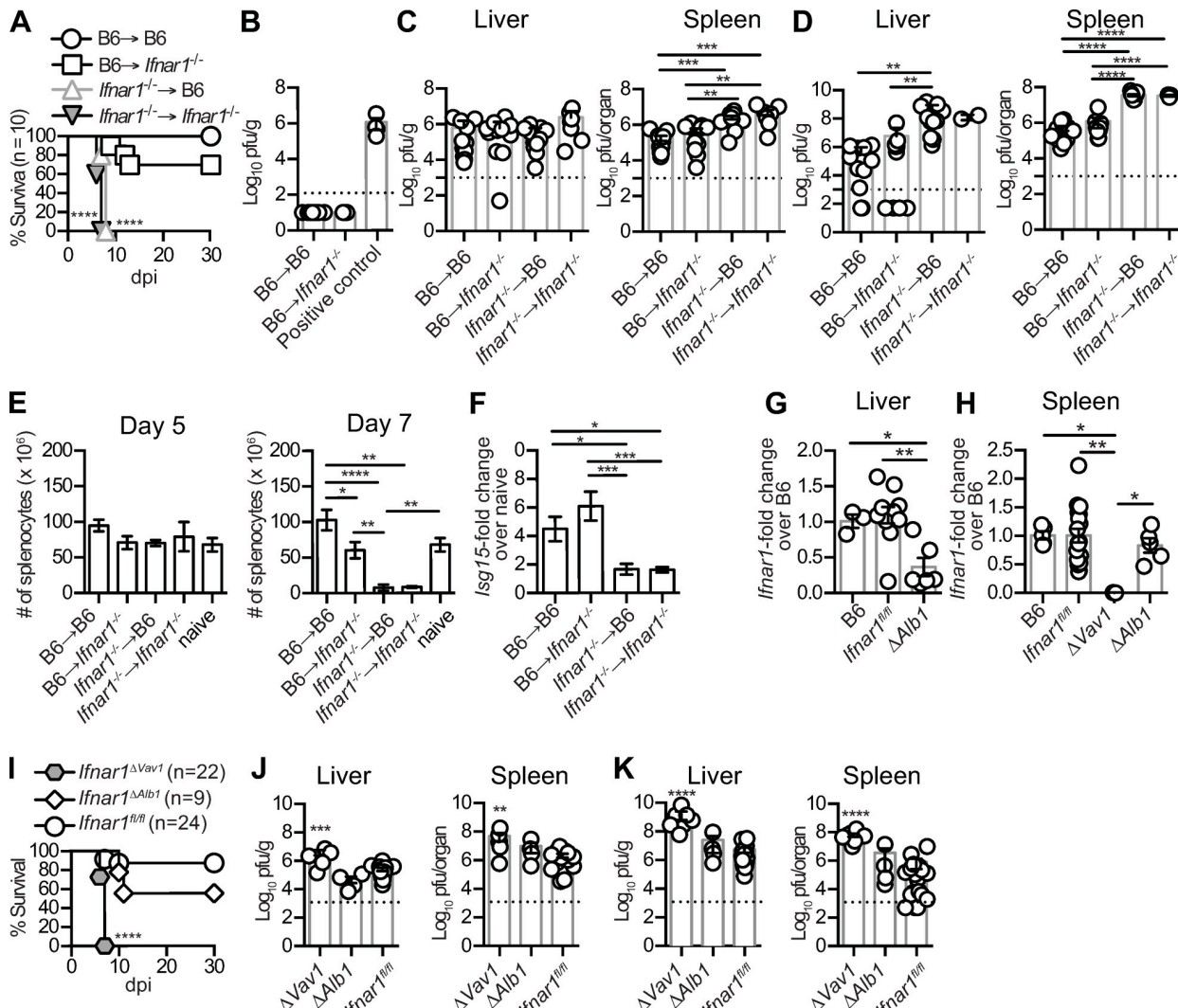

**Fig 1. IFNAR in hematopoietic cells is necessary and sufficient for resistance to lethal mousepox. (A)** Survival of the indicated BMCs infected with 3000 pfu of ECTV-GFP in the footpad. Data were pooled from two independent experiments (Log-rank Mantel-Cox compared to B6→ B6 control group). **(B)** ECTV titers in livers of survivors at endpoint (30 dpi) quantified by plaque assay. The dashed line indicates the detection limit. Positive control: liver sample harvested from infected *Ifnar1*[-/-] mice at 5 dpi. **(C-D)** ECTV titers in the livers and spleens of each BMC group were quantified by plaque assay at 5 **(C)** and 7 dpi **(D)**. The dashed line indicates the detection limit. **(E)** Number of live splenocytes in BMCs naïve or at 5 and 7 dpi with ECTV. Data are represented as mean ± SEM from three pooled independent experiments (N = 11 per group, multiple comparison ANOVA with Tukey correction). **(F)** *Isg15* transcripts in the spleen at 5 dpi quantified by qPCR, normalized to *Gapdh*, and adjusted to fold-change to naïve. Data are represented as mean ± SEM from two independent experiments (N = 6 per group, multiple comparison ANOVA with Tukey correction). **(G-H)** Cre-driven excision of *Ifnar1* gene exon10 in the livers **(G)** and the spleens **(H)** of the indicated naïve mice measured by qPCR, normalized by *Gapdh* transcription, and adjusted to fold-change to B6. **(I)** Survival of the indicated mice infected as in **(A)**. Data pooled from two independent experiments (Log-rank Mantel-Cox compared to *Ifnar1*[fl/fl] mice). **(J-K)** ECTV titers in the livers and spleens of indicated mice at 5 **(J)** and 7 dpi **(K)** quantified as in **(C-D)**. Data are represented as mean ± SEM from two or three pooled independent experiments (N = 8–9 for each Cre[+] mice and N = 12–18 for *Ifnar1*[fl/fl] mice, ANOVA with Tukey correction compared to *Ifnar1*[fl/fl] mice).

*Ifnar1*[-/-] mice survived. The finding that most B6→ *Ifnar1*[-/-] mice were resistant to mousepox was surprising because the radioresistant cell compartment includes hepatocytes, which are an important target of ECTV [16], and epidermal Langerhans cells (LCs), which are required for optimal innate immune responses to ECTV in the draining lymph node (dLN) and resistance to mousepox [17]. Thus, we also tested whether the surviving B6→ *Ifnar1*[-/-] mice had a viral

clearance defect. However, similar to B6→ B6 mice, B6→ $Ifnar1^{-/-}$ survivors had no detectable ECTV in the liver at 30 days post-infection (dpi) (**Fig 1B**).

We also found that compared to B6→ B6 and B6→ $Ifnar1^{-/-}$ mice, the viral loads in $Ifnar1^{-/-}$→ B6 and $Ifnar1^{-/-}$→ $Ifnar1^{-/-}$ mice were increased 10-fold at 5 days post-infection (dpi) in the spleen (**Fig 1C**) and 100-fold in the liver and the spleen at 7 dpi (**Fig 1D**). The uncontrolled viral replication observed in $Ifnar1^{-/-}$→ B6 and $Ifnar1^{-/-}$→ $Ifnar1^{-/-}$ mice correlated with high pathogenicity in the spleen with a 50 to 100-fold reduction in the number of splenocytes at 7 dpi when compared to B6→ B6 and B6→ $Ifnar1^{-/-}$ mice (**Fig 1E**). The presence of IFNAR in the hematopoietic compartment was critical for the induction of ISGs because at 5 dpi $Isg15$ mRNA, as a readout for IFN-I activity, increased to similar levels in the spleens of B6→ B6 and B6→ $Ifnar1^{-/-}$ mice, whereas there was marginal or no $Isg15$ upregulation in the spleens of $Ifnar1^{-/-}$→ B6 and $Ifnar1^{-/-}$→ $Ifnar1^{-/-}$ mice (**Fig 1F**) at a timepoint when numbers of splenocytes were similar among all analyzed groups (**Fig 1E**). The low upregulation of $Isg15$ in $Ifnar1^{-/-}$→ B6 mice also indicates that irradiation successfully eliminated most IFN-responding cells in the spleens of $Ifnar1^{-/-}$→ B6 mice.

To confirm our findings in another system, we crossed $Ifnar1^{fl/fl}$ mice [18] with mice expressing Cre recombinase from the $Vav1$ promoter [19] to delete IFNAR in all hematopoietic cells ($Ifnar1^{\Delta Vav1}$ mice) or from the $Alb$ promoter [20] to delete IFNAR exclusively in hepatocytes ($Ifnar1^{\Delta Alb1}$ mice). The specificity of the deletion was confirmed by quantitative PCR (qPCR) for $Ifnar1$ transcripts in the liver (**Fig 1G**) and the spleen (**Fig 1H**). In agreement with the experiments with BMC, all infected $Ifnar1^{\Delta Vav1}$ mice succumbed rapidly to infection while most $Ifnar1^{\Delta Alb1}$ and $Ifnar1^{fl/fl}$ littermates survived (**Fig 1I**). Moreover, when compared to $Ifnar1^{fl/fl}$ littermates, $Ifnar1^{\Delta Vav1}$ but not $Ifnar1^{\Delta Alb1}$ mice had a significant increase in viral loads in the livers and spleens at 5 (**Fig 1J**) and 7 dpi (**Fig 1K**). Based on these results, we conclude that IFNAR in hematopoietic cells is essential and generally sufficient for ECTV control and resistance to lethal mousepox, but is mostly dispensable in hepatocytes, which are critical targets of ECTV.

## IFNAR in hematopoietic cells is necessary for efficient T cell responses to ECTV

Resistance to mousepox requires functional T and B cells [21–25]. Thus, we analyzed the T cell responses in B6/$Ifnar1^{-/-}$ BMC. Notably, $Ifnar1^{-/-}$→ B6 and $Ifnar1^{-/-}$→ $Ifnar1^{-/-}$ had a significant reduction in the frequency (>10 fold) and absolute numbers (100- to 1000-fold) of CD8 and CD4 T cells expressing granzyme B (GzmB), a marker of activation and cytotoxic potential (**Fig 2A and 2B**), and CD44, a marker of antigen-experienced T cells (**Fig 2A and 2C**) when compared to B6→ B6 mice. In contrast, in B6→ $Ifnar1^{-/-}$ mice, the CD8 T cell responses were not significantly affected, and those of CD4 T cells were significantly but only slightly reduced. This indicates that IFNAR in hematopoietic cells is necessary for efficient T cell responses to ECTV.

## Intrinsic IFNAR on adaptive lymphocytes is dispensable for efficient T cell responses and resistance to mousepox

The data above indicated that IFNAR is required in hematopoietic cells for T cell responses to ECTV but did not indicate whether this requirement is intrinsic or extrinsic for T cells. To test for these possibilities, we adoptively transferred CFSE-labeled WT F1 (B6.CD45.1 x B6. CD45.2) and $Ifnar1^{-/-}$ (CD45.2) cells into IFNAR-sufficient mice (B6.CD45.1). Subsequently, we infected them with ECTV in the footpad and harvested their spleens at 8 dpi (**Fig 3A**). In this context, where the host efficiently controls virus replication and spread, $Ifnar1^{-/-}$ and WT

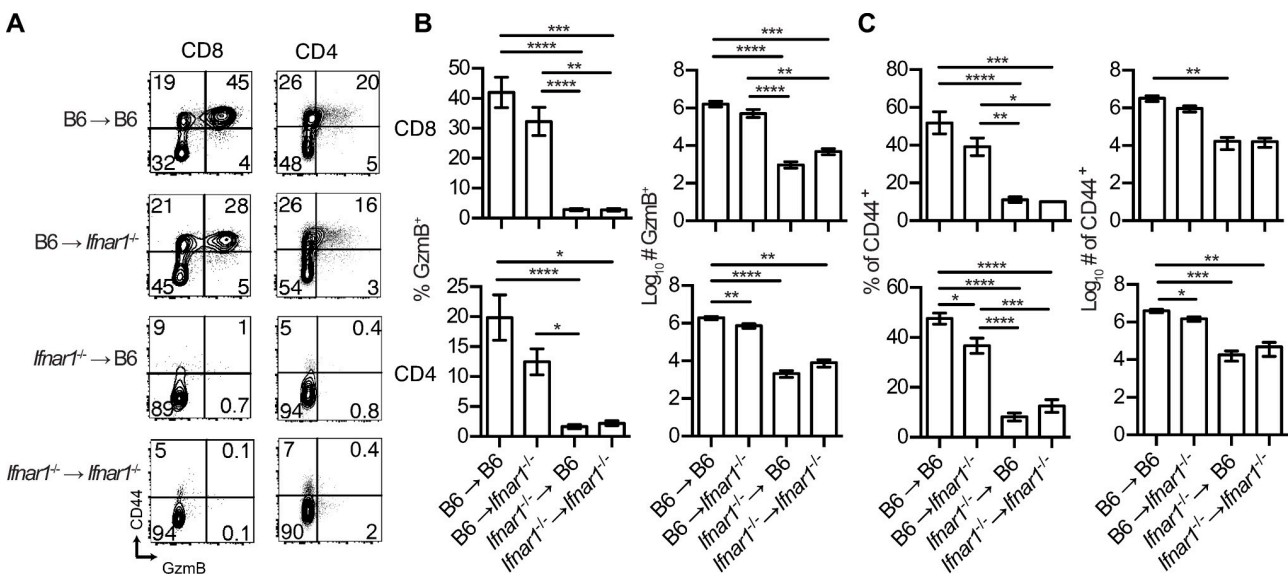

**Fig 2. IFNAR in hematopoietic cells is necessary for efficient T cell responses to ECTV.** BMC mice were infected as in [Fig 1A](), and spleens were harvested at 7 dpi. **(A)** Representative flow cytometry plots showing the frequency of CD44+ and GzmB+ cells on gated CD8 T cells (NK1.1⁻TCRβ+CD8+) and on CD4 T cells (NK1.1⁻TCRβ+CD4+). **(B)** Percentages and numbers of cytotoxic GzmB+ CD8 and CD4 T cells. **(C)** Percentages and numbers of antigen experienced CD44+ CD8 and CD4 T cells. Data are represented as mean ± SEM from three pooled independent experiments (N = 11 per group, N = 3 for *Ifnar1⁻/⁻ → Ifnar1⁻/⁻*, multiple comparison ANOVA with Tukey correction).

CD8 and CD4 T cells responded similarly to the infection as determined by CFSE dilution (**Fig 3B and 3C**). The frequencies of cells expressing CD44 or GzmB (**Fig 3D and 3E**), and the expansion of CD8 T cells specific for the poxvirus immunodominant Kᵇ-restricted epitope TSYKFESV [26] were also similar (**Fig 3D and 3F**).

The experiments above indicated that IFNAR deficient T cells could expand and become effectors in response to ECTV but did not address whether they can protect from mousepox. To address this issue, we irradiated *Rag1⁻/⁻* mice, which lack T and B cells, and reconstituted them with: 1) Bone marrow cells from *Ifnar1⁻/⁻* mice (*Ifnar1⁻/⁻ → Rag1⁻/⁻*) where all hematopoietic cells lacked IFNAR. 2) A mixture of 80% *Rag1⁻/⁻* and 20% *Ifnar1⁻/⁻* bone marrow cells (*Rag1⁻/⁻ + Ifnar1⁻/⁻ → Rag1⁻/⁻* mice) [27], where T cells and B cells lacked IFNAR while most other hematopoietic cells were IFNAR sufficient. 3) 80% *Rag1⁻/⁻* and 20% B6 bone marrow cells (*Rag1⁻/⁻ + B6 → Rag1⁻/⁻*) where all hematopoietic cells were IFNAR sufficient. After challenge with ECTV, all *Ifnar1⁻/⁻ → Rag1⁻/⁻* mice quickly succumbed to infection whereas similar to *Rag1⁻/⁻ + B6 → Rag1⁻/⁻*, most *Rag1⁻/⁻ + Ifnar1⁻/⁻ → Rag1⁻/⁻* mice survived the infection (**Fig 3G**) with mild or no signs of disease. Also, at 30 dpi, they had cleared the virus (**Fig 3H**). Notably, at 22 dpi, surviving *Rag1⁻/⁻ + Ifnar1⁻/⁻ → Rag1⁻/⁻* and *Rag1⁻/⁻ + B6 → Rag1⁻/⁻* mice had similar frequencies of antigen-experienced CD44+ CD8 and CD4 cells and ECTV-specific Kᵇ-TSYKFESV+ CD8 T cells in their blood (**Fig 3I–3K**). Together, these data demonstrate that intrinsic type I IFN stimulation is not necessary for protective T cell responses to mousepox and that the lack of T cell responses observed in *Ifnar1⁻/⁻ → B6* and *Ifnar1⁻/⁻ → Ifnar1⁻/⁻* mice is T cell extrinsic.

## IFNAR on NK cells is essential for resistance to mousepox

Resistance to mousepox requires NK cell cytolytic activity [28–31]. To test whether NK cell antiviral function requires intrinsic IFNAR, we crossed mice expressing Cre as a knock-in in the *Ncr1* locus (NKp46ᶜʳᵉ) with *Ifnar1ᶠˡ/ᶠˡ* mice to generate *Ifnar1ᐞNKp46* mice, which directs

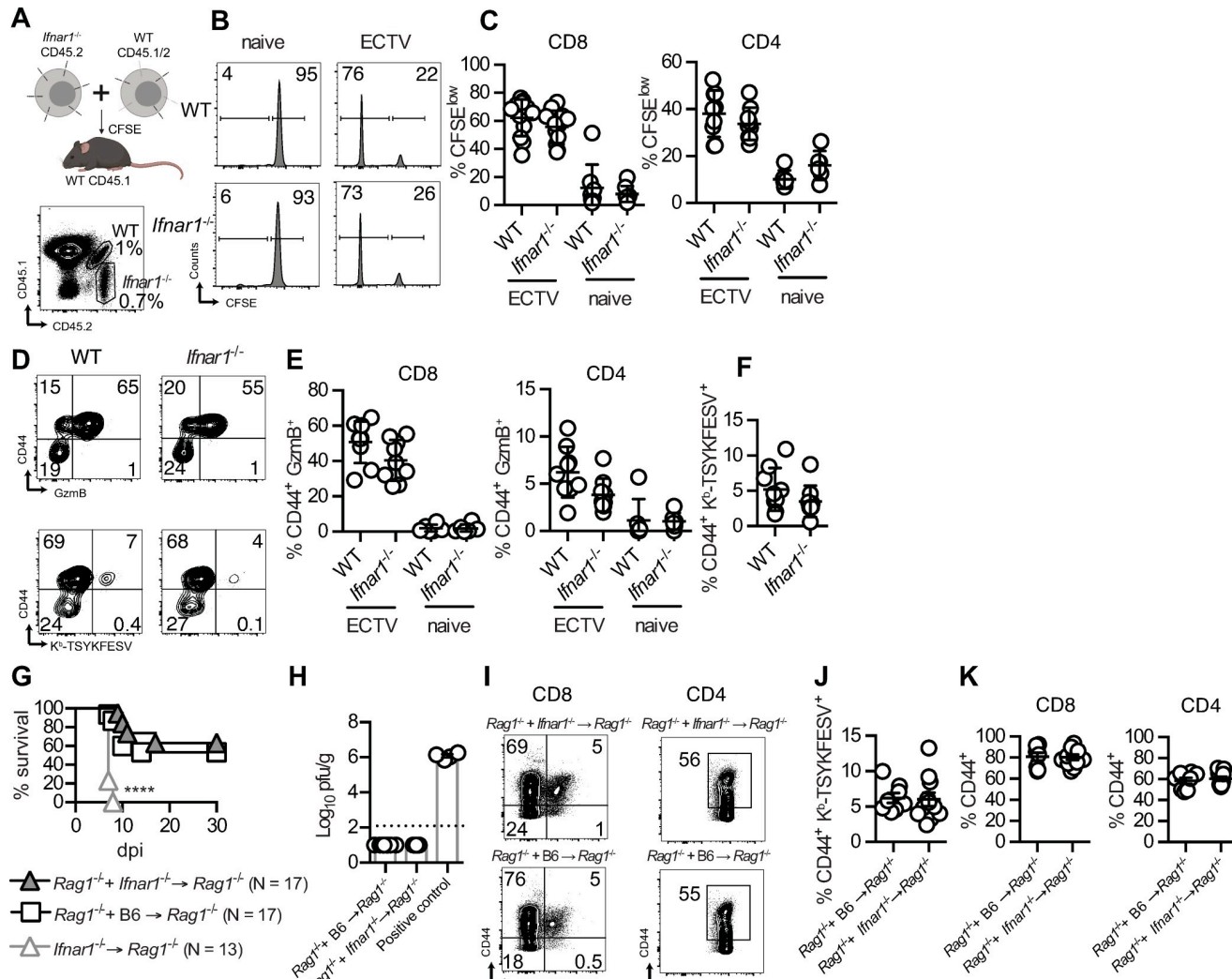

**Fig 3. Intrinsic IFNAR on adaptive lymphocytes is dispensable for efficient T cell responses and resistance to mousepox. (A)** CFSE-labeled WT (F1 [B6.CD45.1 x B6.CD45.2) and *Ifnar1*[-/-] (CD45.2) splenocytes mixed in 1:1 ratio were adoptively transferred i.v. into naïve or infected mousepox resistant WT hosts (B6.CD45.1), and spleens were harvested at 7.5 days following ECTV infection and transfer. Created with BioRender.com. **(B)** Representative flow cytometry histograms of CFSE dilution in gated CD8 T cells (NK1.1[-]TCRβ[+]CD8[+]) in infected and naïve hosts. **(C)** Percentages of CFSE[low] CD8 and CD4 T cells (NK1.1[-]TCRβ[+]CD4[+]). **(D)** Representative flow cytometry plots showing staining with anti-CD44 and -GzmB (top) or anti-CD44 and K[b]-TSYKFESV dimers (bottom) of donor WT or *Ifnar1*[-/-] CD8 T cells. **(E)** Percentages of CD44[+] GzmB[+] CD8 and CD4 T cells. **(F)** Percentages of CD44[+] K[b]-TSYKFESV[+] CD8 T cells. Data are represented as mean ± SEM from two pooled independent experiments (N = 9 for infected hosts and N = 6 for naïve hosts). **(G)** Survival of indicated BMC infected with 3000 pfu of ECTV-GFP in the footpad (Log-rank Mantel-Cox compared to *Rag1*[-/-] + *Ifnar1*[-/-] → *Rag1*[-/-] and to *Rag1*[-/-] + B6 → *Rag1*[-/-]). **(H)** ECTV viral titers in the livers of survivors at endpoint (30 dpi) quantified by plaque assay. The dashed line indicates the detection limit. Positive control: liver sample harvested from infected *Ifnar1*[-/-] mice at 5 dpi. **(I)** Concatenated flow cytometry plots showing levels of K[b]-TSYKFESV[+] CD8 T cells and levels of CD44 expression in CD8 and CD4 T cells in the blood of survivor mice at 22 dpi. The graphs of the left are gated on CD8 T cells. **(J)** Percentages of CD44[+] K[b]-TSYKFESV[+] CD8 T cells. **(K)** Percentages of antigen experienced CD44[+] CD8 and CD4 T cells. Data are represented as mean ± SEM from three pooled independent experiments (N = 8 for *Rag1*[-/-] + B6 → *Rag1*[-/-] and N = 11 for *Rag1*[-/-] + *Ifnar1*[-/-] → *Rag1*[-/-] survivors).

Cre to NK cells and other group 1 innate lymphoid cells (ILCs). By flow cytometry, anti-IFNAR1 monoclonal antibody (mAb) stained T cells and NK cells in *Ifnar1*[fl/fl] mice, T cells but not NK cells in *Ifnar1*[ΔNKp46] mice, and neither T cells nor NK cells in *Ifnar1*[ΔVav1] mice (**Fig 4A**), demonstrating the NK cell-specific and widespread deficiency of IFNAR1 in *Ifnar1*[ΔNKp46] and *Ifnar1*[ΔVav1] mice, respectively. After ECTV infection, most *Ifnar1*[ΔNKp46] mice succumbed to mousepox (**Fig 4B**), with a median survival time of 9.5 days, while most *Ifnar1*[fl/fl] littermates

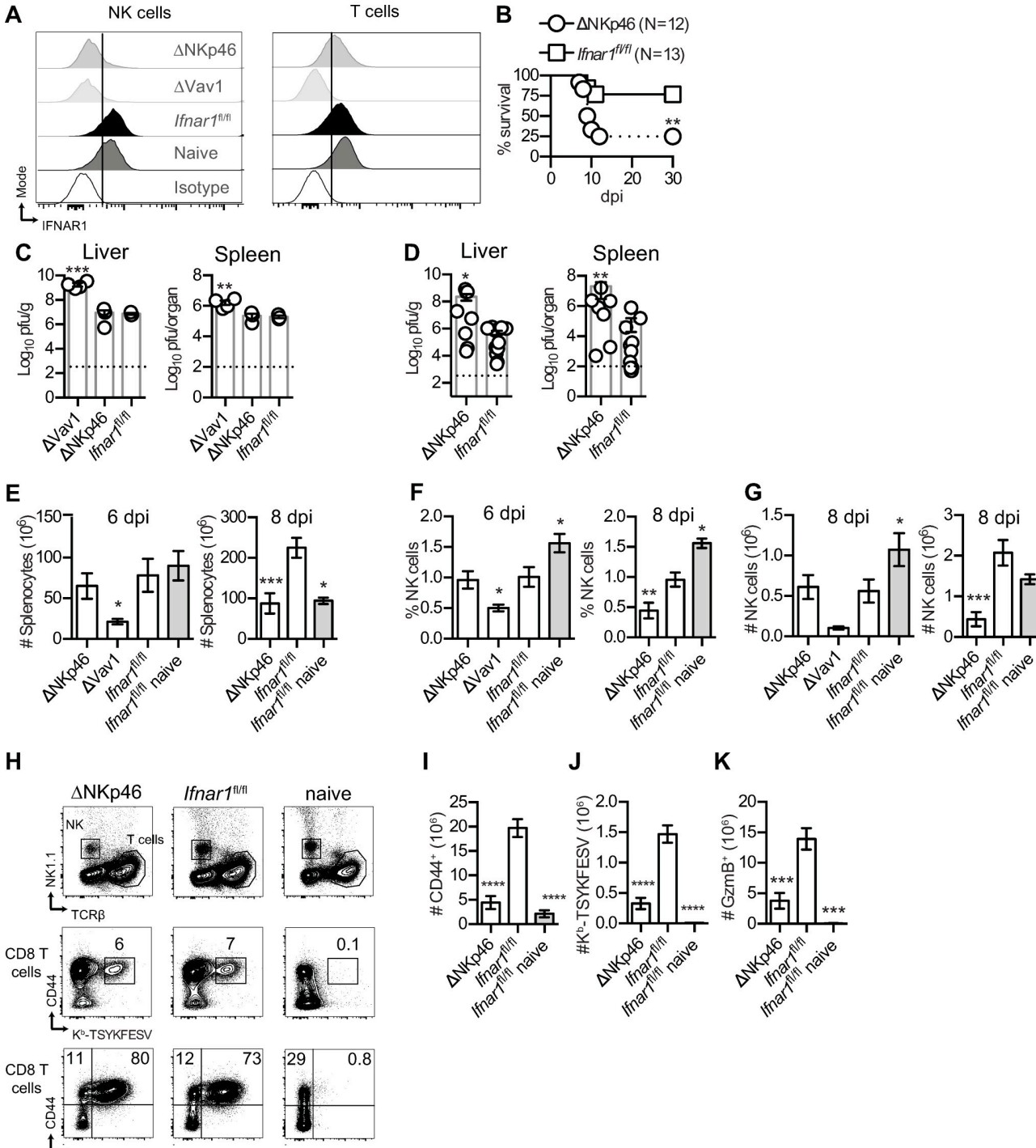

**Fig 4. IFNAR on NK cells is essential for resistance to mousepox (A)** Concatenated histograms showing IFNAR1 expression in gated NK (NK1.1+TCRβ-) or T (NK1.1- TCRβ+) cells from spleens of the indicated infected (6 dpi) mice or of *Ifnar1*fl/fl naïve mice. Concatenated histograms are derived from one independent experiment (N = 3 or four mice for each group). **(B)** Survival of the indicated mice infected with 3000 pfu of ECTV-GFP in the footpad. Data pooled from two independent experiments (Log-rank Mantel-Cox compared to *Ifnar1*fl/fl mice). **(C-D)** ECTV titers the livers and spleens of the indicated mice were quantified by plaque assay at six **(C)** and eight dpi **(D)**. The dashed line indicates the detection limit. **(E-G)** Number of live splenocytes **(E)**, frequency of NK cells **(F)** and numbers of NK cells **(G)** in the spleens of naïve and indicated infected mice at 6 and 8 dpi. **(H)** Concatenated flow cytometry plots showing NK and T cells gating and CD44 expression, Kb-TSYKFESV-specific and GzmB expression in CD8 T cells from spleens of *Ifnar1*fl/fl naïve mice and indicated infected mice at 8 dpi. **(I-K)** Number of antigen experience CD44+ **(I)**, of CD44+ Kb-TSYKFESV+ **(J)** and of CD44+ GzmB+ **(K)** CD8 T cells in the indicated mice at 8 dpi. Data are represented as mean ± SEM from two pooled independent experiments (N = 9 or 10 for each group, ANOVA with Tukey correction compared to *Ifnar1*fl/fl infected mice).

survived, indicating that intrinsic IFNAR signaling in NK cells is vital for optimal resistance to mousepox.

To understand how the lack of intrinsic IFNAR signaling in NK cells predisposes mice to ECTV lethality, *Ifnar1*$^{\Delta NKp46}$, *Ifnar1*$^{\Delta Vav1}$, and *Ifnar1*$^{fl/fl}$ mice were infected with ECTV and analyzed at 6 and 8 dpi. Viral loads in the liver and spleen of *Ifnar1*$^{\Delta NKp46}$ and *Ifnar1*$^{fl/fl}$ mice were similar and significantly lower than in *Ifnar1*$^{\Delta Vav1}$ mice, suggesting that during the first 6 dpi, intrinsic IFNAR signaling in NK cells may not be essential to control ECTV but could be necessary in other hematopoietic cells (**Fig 4C**). The NK cells' intrinsic IFNAR defect resulted in increased viral loads in the spleen and the liver at 8 dpi (**Fig 4D**). Compared to infected *Ifnar1*$^{fl/fl}$ mice, the numbers of splenocytes were significantly reduced in ECTV-infected *Ifnar1*$^{\Delta Vav1}$ but not in *Ifnar1*$^{\Delta NKp46}$ at 6 dpi, but at day 8 dpi, *Ifnar1*$^{\Delta NKp46}$ had reduced numbers of live splenocytes compared to *Ifnar1*$^{fl/fl}$ mice (**Fig 4E**). Notably, at 6 dpi, the frequency and absolute numbers of NK cells were significantly lower in the spleens of *Ifnar1*$^{\Delta Vav1}$ compared to *Ifnar1*$^{\Delta NKp46}$ and *Ifnar1*$^{fl/fl}$ mice, while at 8 but not 6 dpi, there were significant decreases in the frequency and numbers of NK cells in *Ifnar1*$^{\Delta NKp46}$ mice compared to *Ifnar1*$^{fl/fl}$ mice (**Fig 4F and 4G**).

Considering that the mean time to death in *Ifnar1*$^{\Delta NKp46}$ mice infected with ECTV was 9.5 days, we analyzed the T cell response at 8 dpi in the spleens of these mice as a sign of pathogenicity. We found that the frequencies of antigen-experienced (CD44$^+$), ECTV-specific CD8 (CD44$^+$K$^b$- TSYKFESV) and activated cytotoxic CD8 and CD4 (CD44$^+$GzmB$^+$) T cells were similar for *Ifnar1*$^{\Delta NKp46}$ and *Ifnar1*$^{fl/fl}$ mice (**Fig 4H**), suggesting that, as expected, IFNAR deficiency in NK cells does not directly compromise T cell activation *in vivo*. However, given the reduced numbers of live splenocytes in ECTV-infected *Ifnar1*$^{\Delta NKp46}$ mice at 8 dpi, the total number of antigen-experienced CD8 T cells (**Fig 4I**), of TSYKFESV-specific CD8 (**Fig 4J**), and GzmB$^+$ CD8 T cells (**Fig 4K**) was reduced in *Ifnar1*$^{\Delta NKp46}$ mice compared to *Ifnar1*$^{fl/fl}$ mice.

## NK cells require IFNAR for optimal maturation, activation, and cytolytic killing

ECTV-infected *Ifnar1*$^{fl/fl}$ and *Ifnar1*$^{\Delta NKp46}$ mice had no differences in frequency of NK cells that expressed the activation marker CD69, the receptors NKG2D and CD94, which are required to resist to mousepox [28,29] (**S1A Fig**), or the maturation marker KLRG1 [32] (**S1A Fig**). There were also no differences in the mean fluorescence intensity (MFI) of NK cells that expressed the non-classical Major Histocompatibility Class I (MHC-I) molecule Qa-1$^b$, which has been suggested to increase NK cell survival [33] (**S1A Fig**). However, we observed higher frequencies of immature CD27$^+$CD11b$^-$ NK cells [34] in *Ifnar1*$^{\Delta NKp46}$ compared to *Ifnar1*$^{fl/fl}$ mice (**S1B Fig**), suggesting a defect in the ability of NK cells to mature in response to ECTV infection in the absence of intrinsic IFNAR signaling.

We have previously shown that NK cells upregulate GzmB after infection with ECTV [28]. We now find that different to T cells, the upregulation of GzmB in NK cells in response to ECTV is partly dependent on intrinsic IFNAR because at 6 dpi a smaller fraction of NK cells upregulated GzmB in *Ifnar1*$^{\Delta NKp46}$ compared to *Ifnar1*$^{fl/fl}$ infected mice (**Fig 5A, left**). Also, using a mAb to perforin (Prf), we found that most NK cells produce intermediate levels of Prf at steady state. At 6 dpi with ECTV, Prf expression was upregulated in NK cells, many of them to high levels. Notably, most NK cells also upregulated Prf in *Ifnar1*$^{\Delta NKp46}$ mice, but several remained at an intermediate stage, similar to naïve *Ifnar1*$^{fl/fl}$ (**Fig 5A, right**). Consequently, at 6 dpi, a significantly lower fraction of NK cells was GzmB$^{high}$ Prf$^{high}$ in *Ifnar1*$^{\Delta NKp46}$ compared to *Ifnar1*$^{fl/fl}$ mice (**Fig 5B and 5C**).

The previous data suggested a possible defect in NK cell cytolytic function in the absence of intrinsic IFNAR signaling. TAP1-deficient splenocytes (*Tap1*$^{-/-}$) adoptively transferred into

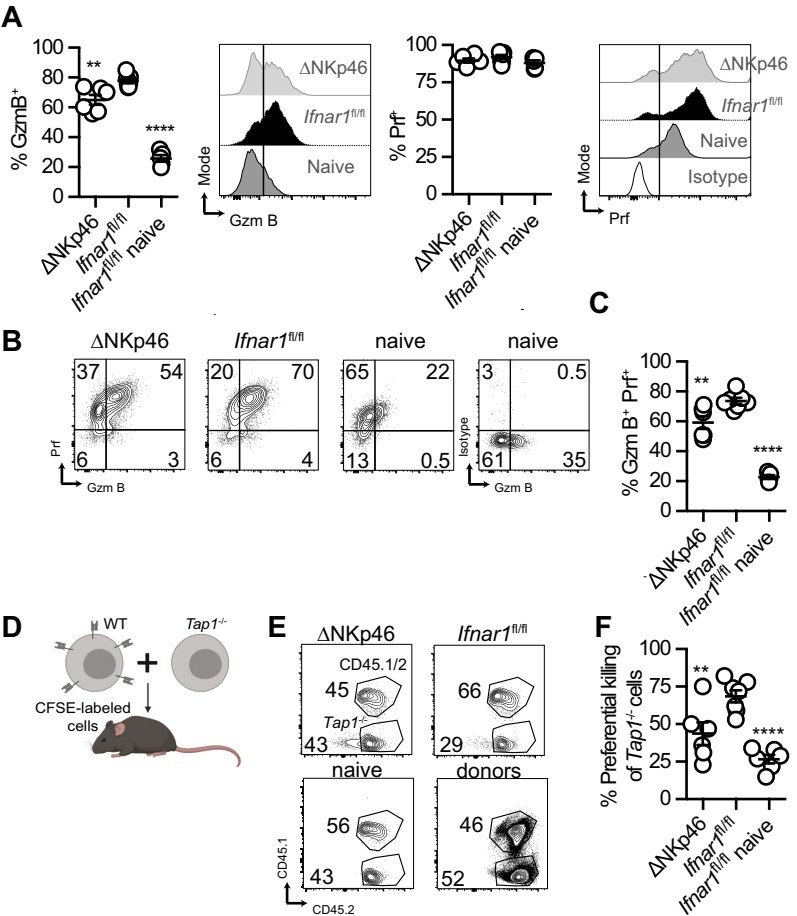

**Fig 5. NK cells require IFNAR for optimal maturation, activation, and cytolytic killing. (A)** Graphs and concatenated flow cytometry histograms depicting GzmB and Prf expression in gated NK cells (NK1.1$^+$ TCRβ$^-$) from spleens of the indicated naïve or infected mice at six dpi. **(B-C)** Concatenated flow cytometry plots **(B)** and frequency **(C)** of GzmB$^+$ Prf$^+$ NK cells (NK1.1$^+$ TCRβ$^-$) in spleens of the indicated mice at 6 dpi. **(D)** Mice were infected with 3000 pfu of WT ECTV in the footpad and CFSE-labeled WT + *Tap1*$^{-/-}$ splenocytes at a 1:1 ratio were transferred i.v. Two hours before organ harvest at 6 dpi. Created with BioRender.com. **(E-F)** Concatenated flow cytometry plots **(E)** and proportions **(F)** of the preferential killing of *Tap1*$^{-/-}$ cells (CFSE$^{high}$ CD45.2) over WT cells (CFSE$^{high}$ B6.CD45.1) normalized to donor transfer ratio in spleens of the indicated naïve and infected mice at 6 dpi. Data are represented as mean ± SEM from two pooled independent experiments (N = 6 or 7 for each group, ANOVA with Tukey correction compared to *Ifnar1*$^{fl/fl}$ infected mice).

naïve mice are killed by NK cells due to their low expression of MHC-I at the cell surface [35,36]. We have previously shown that the preferential killing of *Tap1*$^{-/-}$ over WT splenocytes is significantly higher in mice infected with ECTV for 6 days as compared to naïve mice [37]. Notably, we found that NK cells were significantly less effective at killing *Tap1*$^{-/-}$ cells *in vivo* in *Ifnar1*$^{\Delta NKp46}$ mice than in *Ifnar1*$^{fl/fl}$ mice following ECTV infection (**Fig 5E and 5F**). These results indicate that NK cell intrinsic IFNAR signaling is necessary for optimal NK cell killing during an acute viral infection and required for host protection and survival.

## Intrinsic IFNAR in Lysozyme-expressing myeloid cells is necessary for resistance to mousepox

The data above demonstrates that intrinsic IFNAR is required in NK cells but not in T cells for efficient ECTV control and survival to mousepox. Nevertheless, it did not escape our attention

that $Ifnar1^{\Delta Vav1}$ succumbed faster to ECTV (****P<0.0001) and suffered much higher virus loads than $Ifnar1^{\Delta NKp46}$ mice. This observation suggested that in addition to NK cells, other innate immune cells require intrinsic IFNAR signaling to resist to mousepox.

We have previously shown that iMOs [11] and LCs, but not conventional dendritic cells 1 (cDC1) [17], play vital roles in the development of the innate immune response in the dLN, which is required to restrain early virus spread and to resist lethal mousepox. However, it was unlikely that intrinsic IFNAR signaling was required in LCs since they are radioresistant and B6→$Ifnar1^{-/-}$ mice were resistant to mousepox (**Fig 1A**). MO, macrophages (Mf), and neutrophils express Lysozyme C-2, encoded by the $Lyz2$ gene (ImmGen ULI RNASeq and ImmGen MNP OpenSource). Thus, we crossed $Ifnar1^{fl/fl}$ mice with mice expressing Cre from the $Lyz2$ promoter [38] to generate $Ifnar1^{\Delta Lyz2}$ mice. Consistent with previous reports [39,40], flow cytometric analysis of splenocytes from naïve $Ifnar1^{\Delta Lyz2}$ mice showed that IFNAR was partially deleted in cells of the MO/Mf lineage and also in neutrophils but unaffected in T cells, B cells, NK cells, cDCs, pDCs and eosinophils (**Fig 6A**). When we subdivided MOs according to Ly6C expression, we found that most Ly6C$^{hi}$ but not Ly6C$^{int}$ or Ly6C$^{low}$ MO lost expression of IFNAR in $Ifnar1^{\Delta Lyz2}$ mice (**Fig 6B**). We also attempted to determine IFNAR in subpopulations of cells in the dLN of infected mice, but, unfortunately, anti-IFNAR1 mAb failed to stain any cells in WT or $Ifnar1^{\Delta Lyz2}$ mice, probably due to the high concentration of type I IFNs in the dLN [11] which compete with anti-IFNAR1 (MAR1-5A3) for binding to the extracellular domain of IFNAR [41].

Strikingly, most $Ifnar1^{\Delta Lyz2}$ mice succumbed to ECTV infection with a mean time of death of 8.5 dpi (**Fig 6C**), indicating that IFNAR expression in Lyz2$^+$ cells is necessary for optimal resistance to lethal mousepox. When we compared virus replication between $Ifnar1^{\Delta Lyz2}$ and control $Ifnar1^{fl/fl}$ mice, we found they did not differ in viral loads in the dLN at 1–2 dpi (**Fig 6D**). At 3 dpi, the virus loads in $Ifnar1^{\Delta Lyz2}$ were slightly higher in the dLN (**Fig 6D**), but 100-fold higher in the spleen (**Fig 6E**), indicating faster virus spread. At 5 dpi, $Ifnar1^{\Delta Lyz2}$ had increased virus loads in the liver (**Fig 6F**) and, at 7 dpi, in both liver and spleen (**Fig 6G**). Consistent with the pathological changes usually observed in mice susceptible to mousepox, $Ifnar1^{\Delta Lyz2}$ mice had significant reductions in the number of splenocytes, and numbers of CD44$^+$GzmB$^+$ and CD44$^+$ K$^b$-TSYKFESV$^+$ CD8 T cells in their spleens (**Fig 6H–6J**). Therefore, absence of IFNAR in Lyz2$^+$ cells leads to early viral spread from the dLN to target organs, exacerbates viral replication at later stages of infection, and indirectly affects the strength of the T cell response. Of note, the phenotype in $Ifnar1^{\Delta Lyz2}$ was less intense than in $Ifnar1^{\Delta Vav1}$ mice, suggesting that the roles of IFNAR in Lyz2$^+$, NK cells, and possible other innate immune cells in resistance to mousepox are complementary and additive.

## iMOs require intrinsic IFNAR for optimal transcription of IFN-I genes and for efficient Ly6C and MHC-II expression but not to migrate to dLNs, produce CXCL9, or resist infection

The data above shows that in $Ifnar1^{\Delta Lyz2}$ mice, neutrophils and Ly6C$^+$ MO partially lose IFNAR expression. We and others have previously shown that depletion of phagocytes, including iMOs, render mice susceptible to mousepox [11,21] and that Ly6C$^+$ iMOs play a critical role in the early innate immune response to ECTV in the dLN, which is essential to control early virus spread and resistance to mousepox. On the other hand, neutrophil depletion does not increase susceptibility to ECTV [42]. Furthermore, while Lyz2-cre is known to delete floxed genes in LCs [43] and we have shown that LCs are also critical for the innate immune response to ECTV in the dLN [17], LCs are radioresistant and of host origin in B6$Ifnar1^{-/-}$ mice, which were resistant to mousepox (**Fig 1A**). Thus, it was unlikely that the reason for the

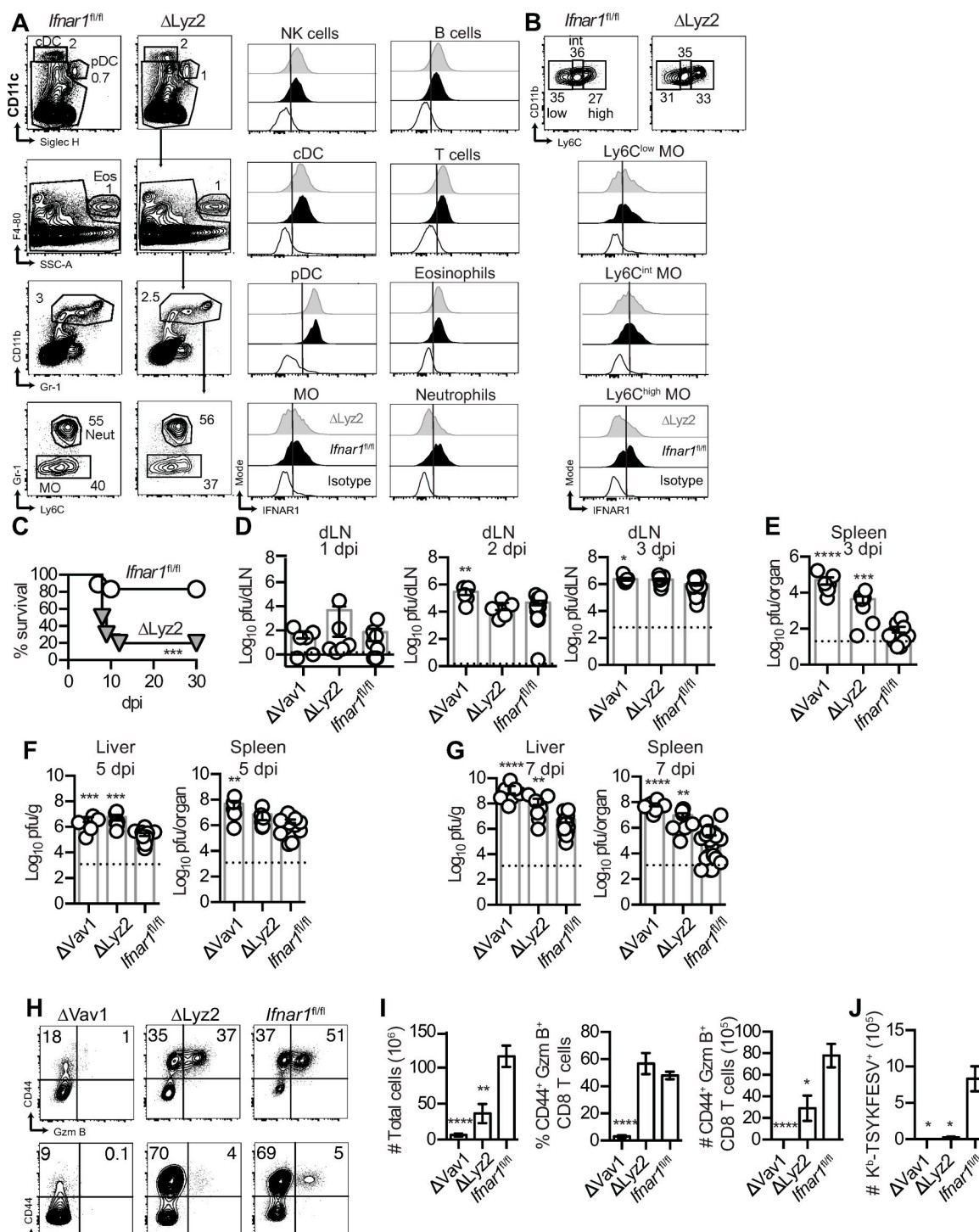

**Fig 6. Intrinsic IFNAR in Lysozyme-expressing myeloid cells is necessary for resistance to mousepox. (A)** Concatenated flow cytometry plots showing the gating strategy for cDCs (CD19⁻TCRβ⁻NK1.1⁻CD11c^high SiglecH⁻), pDCs (CD19⁻TCRβ⁻NK1.1⁻CD11c⁺SiglecH⁺), eosinophils (CD19⁻TCRβ⁻NK1.1⁻SiglecH⁻F4-80⁺SSC-A^high), MO (CD19⁻TCRβ⁻NK1.1⁻SiglecH⁻CD11b⁺Gr-1⁺Ly6C^low/high), and neutrophils (CD19⁻TCRβ⁻NK1.1⁻SiglecH⁻CD11b⁺Gr-1^high) and concatenated histograms showing IFNAR expression in indicated cell populations from spleens of naïve *Ifnar1*^fl/fl and *Ifnar1*^ΔLyz2 mice. **(B)** IFNAR1 expression in Ly6C^low, Ly6C^int, and Ly6C^high gated MO. Histogram legend as in (**A**). Representative data from one out of two independent experiments (N = 6 to 7 for each group). **(C)** Survival of the indicated mice infected s.c. with 3000 pfu of ECTV-GFP. Data pooled from two independent experiments (N = 10 for *Ifnar1*^ΔLyz2 mice and N = 18 for *Ifnar1*^fl/fl mice, Log-rank Mantel-Cox

compared to *Ifnar1*<sup>fl/fl</sup> mice). (**D-E**) ECTV titers in the dLN (**D**) and the spleen (**E**) were quantified by plaque assay at the indicated dpi. The dashed line indicates the detection limit. Data are represented as mean ± SEM from two or three pooled independent experiments (N = 5 to 7 for each Cre<sup>+</sup> mice and N = 11 to 24 for *Ifnar1*<sup>fl/fl</sup> mice; ANOVA with Tukey correction compared to *Ifnar1*<sup>fl/fl</sup> mice). (**F-G**) ECTV titers in the liver and spleen were quantified by plaque assay at 5 (**F**) and 7 dpi (**G**). The dashed line indicates the detection limit. Data are represented as mean ± SEM from two pooled independent experiments (N = 8 to 9 for each Cre<sup>+</sup> mice and N = 18 for *Ifnar1*<sup>fl/fl</sup> mice, ANOVA with Tukey correction compared to *Ifnar1*<sup>fl/fl</sup> mice). (**H**) Representative flow cytometry plots showing levels of CD44 and GzmB expression and levels of K<sup>b</sup>-TSYKFESV-specific CD8 T cells in the spleens of infected mice. (**I**) Total cell numbers, and percentages and total numbers of CD44<sup>+</sup> GzmB<sup>+</sup> CD8 T cells (NK1.1<sup>-</sup> TCRβ<sup>+</sup>CD8<sup>+</sup>) in spleens of the indicated mice at 7 dpi. (**J**) Numbers of CD44<sup>+</sup> K<sup>b</sup>-TSYKFESV-specific CD8 T cells in the spleens of the indicated mice at 7 dpi. Data are represented as mean ± SEM from two pooled independent experiments (N = 8 to 9 for each Cre<sup>+</sup> mice and N = 18 for *Ifnar1*<sup>fl/fl</sup> mice, ANOVA with Tukey correction compared to *Ifnar1*<sup>fl/fl</sup> mice).

susceptibility of *Ifnar1*<sup>ΔLyz2</sup> mice to mousepox was the deletion of *Ifnar1* in neutrophils or LCs. Thus, we next focused our attention on iMOs.

We have previously shown that iMOs are recruited to the dLN at 2–4 dpi and have two critical roles for the control of early systemic virus spread: **1)** uninfected iMOs secrete CXCL9, which promotes NK cell recruitment to the dLN [10] and **2)** infected iMOs produce type I IFNs [11,17]. Thus, we compared the iMOs response in dLN of *Ifnar1*<sup>ΔLyz2</sup> and *Ifnar1*<sup>fl/fl</sup> mice at 2 dpi with ECTV-expressing green fluorescent protein (ECTV-GFP) [28] to distinguish uninfected (GFP<sup>-</sup>) and infected (GFP<sup>+</sup>) cells by flow cytometry. We found that *Ifnar1*<sup>ΔLyz2</sup> and *Ifnar1*<sup>fl/fl</sup> mice did not differ either in the number of iMOs recruited to the dLN (**S2A and S2B Fig**), the frequency of infected iMOs (**S2A and S2C Fig**), the number of uninfected iMOs that produced CXCL9 (**S2A and S2D Fig**) or the frequency and number of NK cells recruited to the dLN by the CXCL9 produced by infected iMOs (**S2A and S2E Fig**). These findings indicate that iMOs do not require intrinsic IFNAR to be recruited to the dLN, resist infection, or recruit NK cells.

Because Ly6C<sup>+</sup> MOs did not fully delete IFNAR in *Ifnar1*<sup>ΔLyz2</sup> mice, we analyzed the accumulation of iMOs at 3 dpi in the dLN of B6.CD45.1 + *Ifnar1*<sup>-/-</sup>→ F1 [B6.CD45.1 x B6.CD45.2] mixed BMC (**S2F Fig**). We observed similar frequencies (**S2F Fig**) and infection rates (**S2G Fig**) of WT and *Ifnar1*<sup>-/-</sup> iMOs, indicating that IFNAR deficiency does not affect iMOs recruitment from the blood to the dLN. However, *Ifnar1*<sup>-/-</sup> iMOs had reduced expression of Ly6C (**S2H Fig**), and, after infection, did not upregulate surface expression of Major Histocompatibility Class II molecules (MHC-II) (**S2I Fig**, **top**). WT but not *Ifnar1*<sup>-/-</sup> infected iMOs also trended to upregulate MHC-I after infection, but the differences were not significant (**S2I Fig**, **bottom**).

Our previous work has shown that ECTV infection induces high levels of IFN-I in the dLN of *Ifnar1*<sup>-/-</sup> mice. However, *Ifnar1*<sup>-/-</sup> mice also endure high virus loads, which may affect the IFN-I response [14]. To test whether intrinsic IFNAR deficiency affects the ability of iMOs to express IFN-I and replicate virus without possible effects of total virus loads in the dLN, we infected B6.CD45.1 + *Ifnar1*<sup>-/-</sup>→ F1 [B6.CD45.1 x B6.CD45.2] mixed BMC with ECTV-GFP and at 3 dpi, we sorted infected (GFP<sup>+</sup>) and uninfected (GFP<sup>-</sup>) WT CD45.1 and *Ifnar1*<sup>-/-</sup> CD45.2 iMOs from dLNs of infected mice (**Fig 7A**). In this context, WT and *Ifnar1*<sup>-/-</sup> iMOs were exposed to the same viral loads within the dLN, and IFN-I production and viral replication was only dependent on IFNAR expression. As baseline controls, we used MOs from the spleens of naïve animals. Using qPCR, we found that compared to naïve MOs, both WT and *Ifnar1*<sup>-/-</sup> infected iMOs upregulated IFN-I genes determined as fold-change over naïve using primers specific for *Ifnb1*, *Ifna4*, and *Ifna5* or for conserved sequences among all *Ifna* genes except *Ifna4* (*Ifna* non-a4) [7]. However, *Ifnar1*<sup>-/-</sup> infected iMOs upregulated the transcription of IFN-I genes to much lower levels than infected WT iMOs (**Fig 7B**). This major reduction in IFN-I gene expression by infected *Ifnar1*<sup>-/-</sup> iMOs suggests the existence of a positive feedback

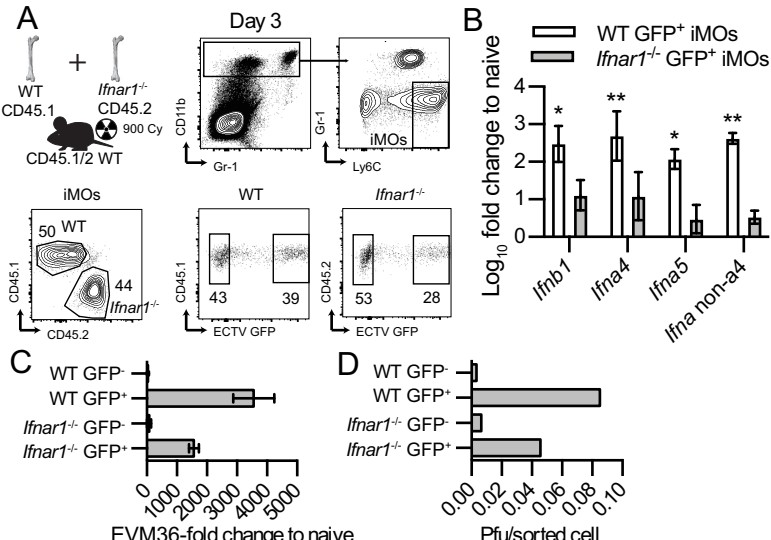

**Fig 7. iMOs require intrinsic IFNAR for optimal transcription of IFN-I genes. (A)** Mixed BMC B6.CD45.1 +*Ifnar1*$^{-/-}$ → F1 [B6.CD45.1 x B6.CD45.2] were infected with 3000 pfu of ECTV-GFP in the footpad and at 3 dpi, uninfected (GFP$^-$) and infected (GFP$^+$) WT and *Ifnar1*$^{-/-}$ iMOs were sorted from the popliteal dLNs as shown in the representative flow cytometry plots. Created with BioRender.com. **(B)** IFN-I transcription quantification was performed by RT-qPCR using RNA extracted from each sorted population as a template. Transcript levels of *Ifnb1*, *Ifna4*, *Ifna5*, and all *Ifna* except *Ifna4* (*Ifna* non-a4) were normalized to *Gapdh*. **(C)** Transcript levels of the ECTV EVM36 gene were quantified as in **(B)**. **(D)** Indicated sorted populations were lysed, and ECTV loads in the indicated iMOs populations were quantified by plaque assay. Data are represented as mean ± SEM from two pooled independent experiments for **(B)** and **(C)** and from one independent experiment for **(D)** (ratio paired Student t-test).

loop for *in vivo* expression of IFN-I which, when impaired, culminates in uncontrolled viral spread from the dLN to target organs and ultimately in death. Notably, sorted infected WT and *Ifnar1*$^{-/-}$ iMOs had similar levels of infection when measured as viral transcripts by qPCR (**Fig 7C**) or as infectious particles/cell by plaque assay (**Fig 7D**). This finding indicates that intrinsic IFNAR signaling does not inhibit viral replication within infected iMOs. Thus, iMOs require IFNAR to for optimal IFN-I gene transcription but not to be directly protected from viral infection.

## Discussion

Here we used a lethal acute mouse model of viral infection to show that IFNAR mediates protection mostly by signaling in hematopoietic cells, as indicated by results obtained with *Ifnar1*$^{-/-}$ BMC (**Fig 1A**) and Cre-LoxP conditional deletions (**Fig 1I**). The observation that IFNAR expression in hepatocytes does not play a significant role in resistance was unexpected, considering that ECTV infects hepatocytes and causes severe hepatitis. In agreement with our observations, others have shown that compared to WT mice, *Ifnar1*$^{\Delta Alb1}$ mice have similar levels of inflammation and virus loads, whereas *Ifnar1*$^{\Delta Lyz2}$ mice have increased viral loads in the liver after intravenous challenge with the closely related poxvirus vaccinia virus [40], suggesting a conserved mechanism for IFNAR protection against poxviruses. We have previously shown that cGAS, which is required for optimal IFN-I production after ECTV infection, is not necessary in parenchymal cells (including hepatocytes) for resistance to mousepox. We now show that resistance to mousepox is also independent of IFNAR in hepatocytes. Whether these characteristics of hepatocytes is the reason for ECTV tropism to the liver is unknown and would be of interest to explore.

Studies using other viral models have shown that IFNAR expression in both hematopoietic and non-hematopoietic cells may contribute to virus control [44–46], suggesting that viral tropism and pathogenicity might determine which cells need IFNAR expression for protection. Notably, it was also previously shown that IFNAR expression in hematopoietic cells is detrimental for host recovery from infection with a mouse-adapted strain of SARS-CoV [47]. However, in this same study, it was also reported that early treatment of susceptible mice with IFN-β was protective and prevented mice from developing lethal pneumonia, suggesting that the timing of IFN-I response might be critical and very early for airborne diseases.

Optimal activation of adaptive lymphocytes requires more than one signal. Signal 1 is provided by the activation of the T or B cell receptor whereas Signal 2 is provided by the activation of co-stimulatory molecules. It has been demonstrated that intrinsic IFNAR expression is essential for optimal T cell function in some contexts [18] suggesting that IFN-I provides a necessary Signal 3 [48]. Accordingly, IFN-I stimulation on T and B cells results in the upregulation of activation markers and cytokine secretion [49,50]. However, we observed that, *in vivo*, *Ifnar1*$^{-/-}$ T and B cells were still capable of protecting the host from a primary ECTV infection and of promoting viral clearance when innate immune cells were IFNAR sufficient (**Fig 3G and 3H**). These findings indicate that T and B cells can exert their full antiviral function independently of IFNAR stimulation and that during a highly inflammatory viral infection such as ECTV, Signal 3 may not be required or may be provided by alternative or redundant mechanisms, such as IL-2, IL-6, IL-12, IL-18 or IFN-γ. In agreement with our observations, it has been shown that conditional deletion of IFNAR in T cells does not affect CD8 T cell responses to murine gammaherpesvirus infection [51]. Moreover, using a mixed BMC approach similar to ours (**Fig 3G**), it has been shown that the adaptive compartment does not require IFNAR signaling for tumor rejection, whereas the innate immune compartment does [27]. Here we did not explore whether intrinsic IFNAR on adaptive lymphocytes is required for the generation of memory CD8 T cells or for protection against secondary infection. However, others have previously shown that *Ifnar1* is not required for resistance to secondary ECTV infection [12].

NK cell function is essential for mousepox resistance, and we have shown here that optimal protection by NK cells requires intrinsic IFNAR (**Fig 4B**). The impaired upregulation of GzmB that we observed in *Ifnar1*$^{-/-}$ NK cells (**Fig 5A**) also occurs during non-lethal infection with mouse cytomegalovirus [33]. It has been shown that intrinsic IFN-I signaling is necessary for optimal NK cell responses to influenza virus and vaccinia virus [52,53], but how this affects virus control was not assessed. Here we observed not only a reduction in GzmB expression but also a decrease in the NK cell pool that expresses both Prf and GzmB (**Fig 5B**). The finding that the preferential killing of *Tap1*$^{-/-}$ cells in ECTV-infected *Ifnar1*$^{\Delta NKp46}$ mice was significantly reduced (**Fig 5E**) highlights the importance of the crosstalk among innate immune cells in the dLN where both iMOs and NK cells migrate to upon infection, and where iMOs express IFN-I. However, IFNAR in iMOs is not required for iMOs or NK cell accumulation in the dLN (**S2A–S2E Fig**).

There were no consistent changes in the expression of various NK cells receptors (**S1A Fig**), but there was an increase in the frequency of immature NK cells in the spleen of infected *Ifnar1*$^{\Delta NKp46}$ mice (**S1B Fig**). Whether this arrest in development contributes to the reduction in NK cell cytotoxicity and decreased resistance to mousepox in *Ifnar1*$^{\Delta NKp46}$ mice is unclear. It is interesting to note that different from NK cells, the upregulation of GzmB in CD8 T cells does not require intrinsic IFNAR (**Fig 3E**). A possible explanation for this difference is that adaptive lymphocytes may rely more on antigen-specific signals for GzmB upregulation, while NK cells, which lack antigen receptors, could be more dependent on direct IFN-I signals.

The finding that partial IFNAR deletion in the monocytic myeloid lineage and neutrophils (**Fig 6A and 6B**) results in a significant increase in susceptibility to lethal mousepox (**Fig 6C**) is intriguing given the ubiquitous expression of IFNAR and the relatively low frequency of myeloid cells in tissues in general. Of note, neutrophil depletion does not increase susceptibility to mousepox [42], cDCs and pDCs express normal levels of IFNAR in *Ifnar1*$^{\Delta Lyz2}$ mice (**Fig 6A**), and LCs and Kupffer cells are of host origin in mousepox resistant B6→*Ifnar1*$^{-/-}$ mice (**Fig 1A**) indicating that none of these cell types are the culprits for the increased susceptibility of *Ifnar1*$^{\Delta Lyz2}$ mice to mousepox.

We have previously shown that infected DCs and iMOs are the primary producers of IFN-I during ECTV infection [11,17,54]. Here, we have used mixed BMCs where WT and *Ifnar1*$^{-/-}$ iMOs were exposed to the same environment during infection to show that IFNAR signaling in infected iMOs is essential to provide a positive feedback loop for IFN-I expression *in vivo* (**Fig 7B**). A positive feedback loop in IFN-I expression was described before *in vitro* using fibroblasts infected with Newcastle disease virus [7]. In that context, virus infection readily induces the transcription of the *Ifnb1* and *Ifna4* genes. The resulting IFN-β and IFN-α4 signal through IFNAR to induce IRF7 expression, which leads to the transcription of non-*Ifna4* genes. Moreover, *Ifnb1*$^{-/-}$ mouse fibroblasts were shown to produce less *Ifna* genes compared to WT fibroblasts after Sendai virus infection [55]. It is unclear whether a similar mechanism also regulates IFN-I production in iMOs *in vivo* or whether different cell types have different regulatory mechanisms. Considering that *Irf7* and *Cgas* are both ISG and necessary for resistance to ECTV [11,14], it is possible that the IFNAR-dependent positive feedback loop of IFN-I might be mediated by the increased expression of these genes. Indeed, mRNA-Seq data of infected and uninfected WT and *Ifnar1*$^{-/-}$ iMOs support this hypothesis and also confirm the IFNAR-dependent positive feedback of IFN-I (manuscript in preparation).

We showed that intrinsic IFNAR signaling does not curb virus transcription (**Fig 7C**) or production of infectious virus (**Fig 7D**) in infected iMOs *in vivo* as a possible mechanism of IFN-I protection from mousepox. Based on this finding, we speculate that the major role of IFN-I signaling in resistance to mousepox is regulating the immune response rather than directly blocking virus replication inside infected cells.

Our data strongly suggest that NK cells and iMOs exert IFNAR-dependent roles that are indispensable for resistance to ECTV infection. We propose that iMOs must sense paracrine or autocrine IFN-I through IFNAR to upregulate IFN-I production, which stimulates NK cells for optimal killing of infected cells and, possibly, other critical protective functions. Our findings contribute to understanding how innate immunity is orchestrated to control viruses and is also of interest in guiding the development of IFN-I-based therapeutic interventions for viral pathogens.

## Materials and methods

### Ethics statement

All the procedures involving mice were carried out in strict accordance with the recommendations in the Eighth Edition of the Guide for the Care and Use of Laboratory Animals of the National Research Council of the National Academies. All experiments were approved by Thomas Jefferson University's Institutional Animal Care and Use Committee under protocol number 01727 "Innate Control of Viral Infections."

### Reagents

All reagents used are listed in Box 1.

**Box 1. List of reagents.**

| REAGENT | SOURCE | IDENTIFIER |
|---|---|---|
| Antibodies | | |
| Hybridoma anti-mouse Fc gamma receptor FcRII, clone 2.4G2 | ATCC | HB-197™ |
| Mouse monoclonal antibody BV785 anti-mouse CD45.1, clone A20 | BioLegend | Cat# 110743, RRID: AB_2563379 |
| Rat monoclonal antibody PE anti-mouse IgG1, clone RMG1-1 | BioLegend | Cat# 406607, RRID: AB_10551439 |
| Mouse monoclonal antibody PerCP-Cy5.5 anti-mouse CD45.2, clone 104 | BioLegend | Cat# 109828, RRID:AB_893350 |
| Mouse monoclonal antibody APC anti-mouse CD45.2, clone 104 | BioLegend | Cat# 109814, RRID:AB_389211 |
| Mouse monoclonal antibody APC anti-mouse NK 1.1, clone PK136 | BioLegend | Cat# 108710, RRID:AB_313397 |
| Mouse monoclonal antibody BV605 anti-mouse NK 1.1, clone PK136 | BioLegend | Cat# 108753, RRID: AB_2686977 |
| Armenian hamster monoclonal antibody BV605 anti-mouse TCRβ, clone H57-597 | BioLegend | Cat# 109241, RRID: AB_2629563 |
| Armenian hamster monoclonal antibody BV785 anti-mouse TCRβ, clone H57-597 | BioLegend | Cat# 109249, RRID: AB_2810347 |
| Rat monoclonal antibody BV711 anti-mouse CD8a, clone 53–6.7 | BioLegend | Cat# 100759, RRID: AB_2563510 |
| Rat monoclonal antibody PE anti-mouse CD8a, clone 53–6.7 | BioLegend | Cat# 100708, RRID:AB_312747 |
| Rat monoclonal antibody APC-Cy7 anti-mouse CD4, clone RM4-5 | BioLegend | Cat# 100526, RRID:AB_312727 |
| Rat monoclonal antibody PE-Cy7 anti-mouse CD4, clone RM4-5 | BioLegend | Cat# 100528 |
| Rat monoclonal antibody BV785 anti-mouse CD4, clone RM4-5 | BioLegend | Cat# 100552, RRID: AB_2563053 |
| Rat monoclonal antibody APC Fire750 anti-mouse CD11b, clone M1/70 | BioLegend | Cat# 101261, RRID: AB_2572121 |
| Rat monoclonal antibody BUV395 anti-mouse CD11b, clone M1/70 | BD Biosciences | Cat# 565976, RRID: AB_2721166 |
| Armenian hamster monoclonal antibody PerCP-Cy5.5 anti-mouse CD27, clone LG.3A10 | BioLegend | Cat# 124214, RRID: AB_2275577 |
| Rat monoclonal antibody BUV395 anti-mouse CD44, clone IM7 | BD Biosciences | Cat# 740215, RRID: AB_2739963 |
| Mouse monoclonal antibody Pacific Blue anti-granzyme B, clone GB11 | BioLegend | Cat# 515408, RRID: AB_2562196 |
| Mouse monoclonal antibody PE anti-mouse perforin, clone S16009A | BioLegend | Cat# 154306, RRID: AB_2721639 |
| Rat monoclonal antibody APC anti-mouse NKG2D, clone CX5 | eBiosciences | Cat# 17-5882-82, RRID: AB_469464 |
| Armenian Hamster monoclonal antibody PE/Cy7 anti-mouse CD69, clone H1.2F3 | BioLegend | Cat# 104511 |
| Mouse monoclonal antibody PE anti-mouse IFNAR1, clone MAR1-5A3 | BioLegend | Cat# 127312, RRID: AB_2248800 |
| Armenian hamster monoclonal antibody PE-Cy7 anti-mouse CD11c, clone N418 | BioLegend | Cat# 117318 |
| Syrian hamster IgG PE-Cy7 anti-mouse KLRG1, clone MAFA, 2F1-Ag | BioLegend | Cat# 138416 |
| Rat monoclonal antibody PerCP/Cy5.5 anti-mouse I-A/I-E, clone M5/114.15.2 | BioLegend | Cat# 107626, RRID: AB_2191071 |
| Rat monoclonal antibody PE anti-mouse Ly6C, clone HK1.4 | BioLegend | Cat# 128008, RRID: AB_1186132 |
| Rat monoclonal antibody APC anti-mouse Ly6C, clone HK1.4 | BioLegend | Cat# 128016, RRID: AB_1732076 |
| Armenian hamster monoclonal antibody PE anti-mouse CXCL9, clone MIG-2F5.5 | BioLegend | Cat# 515603, RRID: AB_2245490 |

(*Continued*)

**Box 1.** (Continued)

| | | |
|---|---|---|
| Rat monoclonal antibody Pacific Blue anti-mouse Ly6C/Ly6G (Gr-1), clone RB6-8C5 | BioLegend | Cat# 108430, RRID:AB_893556 |
| Rat monoclonal antibody APC Fire750 anti-mouse Ly6C/Ly6G (Gr-1), clone RB6-8C5 | BioLegend | Cat# 108456, RRID: AB_2616737 |
| Rat monoclonal antibody BV785 anti-mouse CD19, clone 6D5 | BioLegend | Cat# 115543, RRID: AB_11218994 |
| Rat monoclonal antibody PE anti-mouse CD94, clone 18d3 | eBiosciences | Cat# 12-0941-81, RRID: AB_465783 |
| Mouse monoclonal antibody PE-Cy7 anti-mouse H-2K$^b$/H-2D$^b$, clone 28-8-6 | BioLegend | Cat# 114616 |
| Rat monoclonal antibody FITC anti-mouse Siglec-H, clone 551 | BioLegend | Cat# 129604, RRID: AB_1227761 |
| Rat monoclonal antibody anti-mouse CD16/CD32, clone 93 | BioLegend | Cat#101302 |
| Rat monoclonal antibody BV605 anti-mouse CD19, clone 6D5 | BioLegend | Cat# 115540, RRID: AB_2563067 |
| Bacterial and Virus Strains | | |
| ECTV Moscow | ATCC | ATCC VR-1374 |
| ECTV-GFP | [28] | N/A |
| Chemicals, Peptides, and Recombinant Proteins | | |
| CellTrace™ CFSE Cell Proliferation Kit | Thermo Fisher Scientific | C34554 |
| DMEM media | CORNING | 10-013-CV |
| RPMI media | CORNING | 10-040-CV |
| Penicillin Streptomycin Solution, 100x | CORNING | 30-002-Cl |
| Fetal Bovine Serum, Heat Inactivated | Seradigm | 1500–500 |
| GlutaMAX 100x | Gibco | 35050–061 |
| Hepes 1M | CORNING | 25-060-Cl |
| BD Cytofix/Cytoperm™ | BD Biosciences | 51-2090KZ |
| Peptide TSYKFESV | Genscript | |
| NA/LE mouse anti-mouse Qa-1$^b$, 6A8.6F10.1A6 | BD Pharmingen | 559827 |
| Dimer K$^b$—BD™ Dimer X | BD Biosciences | 550750 |
| Liberase™ | Roche | 05 401 119 001 |
| RNase-free DNase Set | Qiagen | 79254 |
| Critical Commercial Assays | | |
| cDNA synthesis with High Capacity cDNA Reverse Transcription Kit | Applied Biosystems™ Thermo Fisher Scientific | 4368814 |
| RNeasy® Mini Kit | Qiagen | 74106 |
| iTaq™ Universal SYBR® Green PCR Supermix | BioRad | 172–5124 |
| RNA Clean and Concentrator™-5 | Zymo Research | R1014 |
| Experimental Models: Cell Lines | | |
| Monkey *C. aethiops* epithelial kidney BS-C-1 cells | ATCC | CCL-26 |
| Experimental Models: Organisms/Strains | | |
| Mouse: C57BL/6NCrl | Charles River | 027 |
| Mouse: B6.SJL-*Ptprc$^a$Pepc$^b$*/BoyCrl | Charles River | 494 |
| Mouse: C57BL/6 *Ifnar1$^{-/-}$* mice | Thomas Moran (Mount Sinai School of Medicine, New York, NY) | |
| Mouse: C57BL/6 *Ifnar1$^{fl/fl}$* mice | Ulrich Kalinke (Institute for Experimental Infection Research, Brunswick, Germany) | |
| Mouse: B6.129P2-Lyz2$^{tm1(cre)Ifo}$/J | Jackson Laboratory | JAX: 004781 |
| Mouse: B6.Cg-Commd10$^{Tg(Vav1-icre)A2Kio}$/J | Jackson Laboratory | JAX: 008610 |
| Mouse: B6.Cg-Speer6-ps1$^{Tg(Alb-cre)21Mgn}$/J | Jackson Laboratory | JAX: 003574 |

*(Continued)*

**Box 1.** (Continued)

| | | |
|---|---|---|
| Mouse: B6.129S7-Rag1[tm1Mom]/J | Jackson Laboratory | JAX: 002216 |
| Mouse: C57BL/6-Ncr1tm1.1(iCre)Viv/Orl | E. Vivier (Marseille, France) | |
| Oligonucleotides | | |
| Primer *Gapdh* forward | IDT | tgtccgtcgtggatctgac |
| Primer *Gapdh* reverse | IDT | cctgcttcaccaccttcttg |
| Primer *Ifnb1* forward | IDT | ctggcttccatcatgaacaa |
| Primer *Ifnb1* reverse | IDT | agagggcgtggtggagaa |
| Primer *Ifna4* forward | IDT | tcaagccatccttgtgctaa |
| Primer *Ifna4* reverse | IDT | gtcttttgatgtgaagaggttcaa |
| Primer *Ifna5* forward | IDT | gccttaaccctcctggtaaaa |
| Primer *Ifna5* reverse | IDT | tcctgtgggaatccaaagtc |
| Primer *Ifna* non-a4 forward | IDT | ARSYtgtStgatgcaRcaggt |
| Primer *Ifna* non-a4 reverse | IDT | ggWacacagtgatcctgtgg |
| Primer *Isg15* forward | IDT | agtcgacccagtctctgactct |
| Primer *Isg15* reverse | IDT | ccccagcatcttcacctta |
| Primer EVM36 forward | IDT | tgccagttagcactgcgtat |
| Primer EVM36 reverse | IDT | aggtgttctggagaatcaaaga |
| Primer *Ifnar1* exon 10 forward | IDT | gctttgaggagcgtctggaa |
| Primer *Ifnar1* exon 10 reverse | IDT | tcatcaatactgcggggagg |
| Software and Algorithms | | |
| Prism 8 Software | GraphPad Software | |
| FlowJo™ version 10 | Treestar | |
| Other | | |
| TissueLyzer II | Qiagen | 85300 |
| Thermocycler CFX96 Real-Time System | BioRad | |
| BD FORTESSA™ cytometer | BD Biosciences | |
| FACSAria™ II sorter | BD Biosciences | |

## Mice

All mice used in experiments were 8 to 12 weeks old or 16 to 18 weeks old for BMC. To make BMC, we used 4 to 6 weeks old females. For all other experiments, both males and females were used. No sex differences were observed. C57BL/6NCrl (B6) and B6.SJL-*Ptprc*[a]*Pepc*[b]/BoyCrl (CD45.1) mice were purchased from Charles River directly for experiments or as breeders. B6 and B6.CD45.1 were bred in-house to generate F1 [B6.CD45.1 x B6.CD45.2] heterozygous mice. *Ifnar1*[-/-] mice backcrossed to B6 were gifts from Dr. Thomas Moran (Mount Sinai School of Medicine, New York, NY). *Ifnar1*[fl/fl] mice were provided by Dr. John Wherry (University of Pennsylvania, Philadelphia, PA) by permission from Dr. Ulrich Kalinke (Institute for Experimental Infection Research, Brunswick, Germany). C57BL/6-Ncr1tm1.1(iCre) Viv/Orl mice (Nkp46-Cre) were a gift of Dr. E. Vivier (Marseille, France). B6.129S7-Rag1[tm1-Mom]/J, B6.129P2-Lyz2[tm1(cre)Ifo]/J, B6.Cg-Commd10[Tg(Vav1-icre)A2Kio]/J, B6.Cg-Speer6-ps1[Tg(Alb-cre)21Mgn]/J were purchased from the Jackson Laboratory. Colonies were bred at Thomas Jefferson University under specific pathogen-free conditions.

## Viruses and infection

ECTV Moscow strain (WT) (ATCC VR-1374) and ECTV-GFP [28] were propagated in tissue culture, as previously described [13]. Briefly, BS-C-1 cells grown in DMEM media

supplemented with 10% fetal bovine serum (FBS), 100 IU penicillin, 100 μg/mL streptomycin, 1x GlutaMAX, and 10mM HEPES to confluency were infected at MOI 0.01, and viruses were harvested 4 to 5 days later. For that, cells were rinsed with phosphate-buffered solution (PBS), scraped and concentrated by centrifugation. ECTV was released by multiple freezing and thawing cycles. Viruses stocks were sonicated and titrated by plaque assay in BS-C-1 cells. Mice were infected subcutaneously in the rear footpad with 3000 plaque-forming units (PFU) of ECTV WT or ECTV-GFP as indicated. For sorting monocytes from dLN, mice were infected in both rear footpads with 3000 PFU of ECTV-GFP in each footpad. In survival curves, mice were monitored daily and sacrificed whenever illness resulted in a lack of activity and unresponsiveness to touch. Euthanasia was performed according to the 2013 edition of the AVMA Guideline for the Euthanasia of Animals. For virus titer quantification, entire or portions of spleens and portions of livers were homogenized in supplemented DMEM media using a TissueLyzer II (Qiagen), and organ titers were determined by plaque assay on BS-C-1 cells as described before [13]. Shortly, BS-C-1 cells at confluency were infected with 10-fold dilutions of each virus sample for 2 hours at 37°C. Infected cells were grown in 1% carboxy-methyl cellulose overlay containing DMEM media supplemented with 2.5% FBS, 100 IU/mL penicillin, 100 μg/mL streptomycin, 1x GlutaMAX and 10mM HEPES. Virus plaques were quantified at 5 dpi after staining with 0.2% crystal violet dissolved in 20% methanol.

## Bone marrow chimeras

Were prepared as previously described [56]. Briefly, 4- to 6-week-old mice previously treated for three days with acidified water (pH 2.5) were irradiated with 900 Gy using a GammaCell 40 apparatus (Nordion Inc.). Bone marrow cells were isolated in RPMI media supplemented with 10% FBS, 100 IU/mL penicillin, and 100 μg/mL streptomycin. Red blood cells were lysed with 0.84% $NH_4Cl$ and cells were filtered and counted. Irradiated mice were reconstituted intravenously with 5–10 million bone marrow cells from donors. Chimeras were given acidified water for four weeks and rested for eight weeks after reconstitution.

## Flow cytometry and sorting

Flow cytometry was performed as previously described [13]. Briefly, organs were processed into single-cell suspensions. Spleens were smashed with frosted slides, and cells were washed with RPMI media supplemented with 10% FBS, 100 IU/mL penicillin, and 100 μg/mL streptomycin. Livers were cut into smaller pieces and smashed using syringe plungers and metal strainers. White and red blood cells were percoll-enriched, and hepatocytes were discarded. Red blood cells were lysed with 0.84% $NH_4Cl$, washed in media, and counted for staining. Cells were incubated for 25 min in the fridge in the presence of ab 2.4G2 (Fc gamma receptor FcRII, ATCC) and surface antibodies. $K^b$ dimer was loaded with TSYKFESV peptide overnight at 37°C and subsequently incubated with PE anti-mouse IgG1 for 1 hour at room temperature. $K^b$-TSYKFESV PE IgG1 conjugates were incubated with cells for 1 hour at 4°C. According to the manufacturer's instructions, cells were fixed, permeabilized, and stained for 30 min for intracellular markers with BD Cytofix/Cytoperm™ kit. Data were acquired with a BD FOR-TESSA™ cytometer and analyzed with FlowJo™ version 10. For sorting iMOs from dLN, mice were infected as previously with ECTV-GFP in both rear footpads, and 3 dpi popliteal dLN were collected from infected mice. Spleens from naïve mice were used for sorting naïve iMOs. dLN were treated with Liberase TM (1.67 Wünsch units/mL) for 30 min in PBS supplemented with 10mM Hepes. Single-cell suspensions were prepared with 70μm strainer, and repetitive washes with PBS supplemented with 2% BSA and 10mM Hepes. Cells were counted, treated with anti-mouse CD16/CD32 (BioLegend—0.25μL per million of cells) for 15 min in the at

4˚C in PBS supplemented with 2% BSA and 1mM EDTA and subsequently stained with surface antibodies for 20 min at 4˚C. Cells were washed and sorted with a FACSAria™ II sorter in PBS supplemented with 1% BSA, 25mM Hepes, and 1mM EDTA.

## CFSE labeling and adoptive transfers

Splenocytes from indicated donor mice were isolated in RPMI media supplemented with 10% FBS, 100 IU/mL penicillin, and 100 µg/mL streptomycin. Red blood cells were lysed with 0.84% $NH_4Cl$. Splenocytes were washed with a pre-warmed phosphate-buffered solution (PBS) containing 0.1% bovine serum albumin (BSA) and filtered in 70µm sterile nylon mesh. Cells were adjusted to 50 million cells/ml concentration, and a 1:1 ratio of donor mix was labeled with 4µM CellTrace™ CFSE Cell Proliferation Kit for 10 min at 37˚C. Recipient mice were injected intravenously with 0.4 to 2 x $10^7$ of CFSE-labeled cells and infected at indicated times with ECTV.

## RNA preparation and Reverse Transcription-qPCR

Total RNA from spleens and livers was purified with the RNeasy Mini Kit (Qiagen) and treated with DNase I (Qiagen). RNA quantified using NanoDrop, and 300 ng were used as a template for cDNA synthesis with High Capacity cDNA Reverse Transcription Kit in a 20µL reaction. Quantitative PCR (qPCR) was performed as previously described [11,54]. Shortly, 1µL of cDNA was used as a template for amplification using SYBR Green PCR Master Mix through 92˚C for 3 min, 25 cycles of 92˚C for 30 sec, 56˚C for 45 sec, 65˚C for 50 sec, plate read in a Thermocycler CFX96 Real-Time System (BioRad). For purifying RNA from sorted populations, cells were sorted into Trizol, and the RNA was purified with the RNA Clean and Concentrator (Zymo Research). RNA was eluted in 10µL, and all RNA was used as a template for cDNA synthesis. Gene expression was normalized by *Gapdh* levels.

## Quantification and statistical analysis

Data were analyzed with Prism 8 Software. Log-rank (Mantel-Cox) analysis was used for survival experiments. ANOVA with Tukey correction for multiple comparisons or paired and unpaired Student's t-test were used as applicable for other experiments. In all figures, *p<0.05, **p<0.01, ***p<0.001, ****p<0.0001. Graphs show mean +/- SEM.

## Supporting information

**S1 Fig. (relate to Fig 5). Deletion of IFNAR1 in NK cells results in increased frequency of immature NK cells after ECTV.** (A-B) The indicated mice were infected with 3000 pfu of ECTV-GFP in the footpad and spleens were harvested at 6 dpi. (A) Concatenated histograms and graphs showing proportions of KLRG1, NKG2D, CD69 and CD94 expression in gated NK cells (NK1.1+ TCRβ-) and MFI of expression of Qa-1b+ NK cells. (B) Concatenated flow cytometry plots and pie charts showing average proportions of NK cells subpopulations based on CD27 and CD11b expression. Data are represented as mean from two pooled independent experiments (N = 6 or 7 for each group, ANOVA with Tukey correction compared to *Ifnar1*fl/fl infected mice).
(TIF)

**S2 Fig. (related to Fig 7). iMOs require intrinsic IFNAR for efficient Ly6C and MHC-II expression but not to migrate to dLNs, produce CXCL9, or resist infection. (A-D)** Indicated mice were infected with 3000 pfu of ECTV-GFP in the footpad, and at 2 dpi, the popliteal dLN and collateral non-draining lymph node (ndLN) were harvested. **(A)** Concatenated flow

cytometry plots showing proportions of iMOs (CD8⁻CD4⁻NK1.1⁻CD11b⁺Gr-1⁺), proportions of infected (ECTV⁺), and uninfected (ECTV⁻) iMOs determined by GFP expression, proportions of CXCL9 expression within ECTV⁺ and ECTV⁻ iMOs populations, and proportions of NK cells (CD8⁻CD4⁻NK1.1⁺) observed in the dLN. **(B)** Numbers of iMOs in dLN. **(C)** Frequency of infected iMOs expressing GFP in dLN. **(D)** Numbers of uninfected iMOs expressing CXCL9. **(E)** Frequency and numbers of NK cells in dLN. Data are represented as mean ± SEM from two pooled independent experiments (N = 8–9 for each Cre⁺ mice and N = 15 for *Ifnar1*^fl/fl mice, ANOVA with Tukey correction compared to *Ifnar1*^fl/fl mice). **(F-I)** Mixed BMCs B6.CD45.1 + *Ifnar1*^-/- → F1 [B6.CD45.1 x B6.CD45.2] were infected with 3000 pfu of ECTV-GFP in the footpad and the popliteal dLNs were harvested at 3 dpi. **(F)** Flow cytometry histograms showing iMOs proportions within WT (B6.CD45.1) and *Ifnar1*^-/- (CD45.2) populations and histograms depicting levels of Ly6C expression in each of these iMOs populations. **(G)** Flow cytometry plots showing ECTV⁻ and ECTV⁺ iMOs as determined by GFP expression in WT and *Ifnar1*^-/- iMOs and graph showing the percentages of ECTV⁺ iMOs. **(H)** Flow cytometry histograms showing Ly6C expression in infected and uninfected iMOs and mean fluorescence intensity of Ly6C⁺ in ECTV⁻ and ECTV⁺ iMOs populations. **(I)** Flow cytometry histograms showing levels of K^bD^b MHC-I and MHC-II expression in Ly6C⁺ ECTV⁺ iMOs and mean fluorescence intensity of these molecules in ECTV⁺ and ECTV⁻ Ly6C⁺ iMOs. Data are represented as mean ± SEM from three pooled independent experiments in which dLNs from individual BMCs (N = 8–10 mice per experiment) were pooled together for flow cytometry analysis. The statistical analysis shown is multiple comparison ANOVA with Tukey correction.
(TIF)

## Acknowledgments

We wish to thank the Thomas Jefferson University Flow cytometry and Laboratory Animal core facilities for their invaluable services.

## Author Contributions

**Conceptualization:** Carolina R. Melo-Silva, Luis J. Sigal.

**Data curation:** Carolina R. Melo-Silva.

**Formal analysis:** Carolina R. Melo-Silva, Luis J. Sigal.

**Funding acquisition:** Luis J. Sigal.

**Investigation:** Carolina R. Melo-Silva, Pedro Alves-Peixoto, Natasha Heath, Lingjuan Tang, Brian Montoya, Cory J. Knudson, Colby Stotesbury, Maria Ferez, Eric Wong.

**Methodology:** Carolina R. Melo-Silva.

**Project administration:** Luis J. Sigal.

**Resources:** Luis J. Sigal.

**Supervision:** Luis J. Sigal.

**Writing – original draft:** Carolina R. Melo-Silva.

**Writing – review & editing:** Carolina R. Melo-Silva, Luis J. Sigal.

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
