## [Decision Letter · Decision Letter 0]

5 Feb 2021

Dear Luis,

Thank you very much for submitting your manuscript "Resistance to lethal ectromelia virus infection requires Type I interferon receptor in natural killer cells and monocytes but not in adaptive immune or parenchymal cells" for consideration at PLOS Pathogens. As with all papers reviewed by the journal, your manuscript was reviewed by members of the editorial board and by several independent reviewers. The reviewers appreciated the attention to an important topic. Based on the reviews, we are likely to accept this manuscript for publication, providing that you modify the manuscript according to the review recommendations.

Sincerely,

Gerd Sutter

Guest Editor

PLOS Pathogens

Klaus Früh

Section Editor

PLOS Pathogens

Kasturi Haldar

Editor-in-Chief

PLOS Pathogens

orcid.org/0000-0001-5065-158X

Michael Malim

Editor-in-Chief

PLOS Pathogens

orcid.org/0000-0002-7699-2064

Reviewer Comments (if any, and for reference):

Reviewer's Responses to Questions

**Part I - Summary**

Reviewer #1: This manuscript adresses the requirement of IFN signaling in different immune cell types for resistance to lethal ectromelia virus infection. Whilst the importance of type I IFN in mousepox resistance is well-established, whether all or specialised cellular compartments need IFN-I signalling is unclear, so the biological question underpinning the research is relevant. The findings reported are very interesting and will help us understand how the host immune response is executed and compartimentalised during viral infection, an area to which the senior author of the manuscript has substantially contributed. I have a number of questions/clarifications, two of which (in the major issue section) should be addressed in order to strenghten the study.

Reviewer #2: Melo-Silva et al. revealed by experiments with bone marrow chimeric mice that IFNAR signaling of hematopoitic cells is needed to control mousepox infection. Despite mousepox infection induces hepatitis, IFNAR signaling of hepatocytes did not crucially contribute to protection. Furthermore, IFNAR signaling of T and B cells was not needed for protective T cell responses, whereas intrinsic IFNAR signaling of NK cells as well as of inflammatory monocytes was vital for optimal resistance.

This study shows important new and surprising data of how type I IFN confers resistance to mousepox infection. In this study experiments with classical bone marrow chimeric mice and more advanced conditional knock out mice were combined to elegantly work out the specific role of NK cells and inflammatory monocytes. The experiments were thoughtfully planned carefully carried out. It is appreciated that the authors performed many important control experiments such as the FACS analysis of IFNAR expression in IFNAR1deltaNKp46, IFNAR1deltaVav1 and IFNAR1fl/fl mice. The manuscript is very well written.

Reviewer #3: The viral pathogen ectromelia virus ECTV is causing mousepox, a disease which is of variable clinical outcome depending on the strain of mice used. C57BL/6 mice are naturally resistant and survive without disease, while broadly IFNAR-deficient B6 mice rapidly die. In this manuscript by Melo-Silva et al., the authors expand on this finding and explore the effect of tissue-specific IFNAR-deficiency on survival in the ECTV foodpad infection model. They elegantly show that IFNAR expression on NK cells and monocytes, more specifically iMOs, is required for protection, while IFNAR signalling in T cells seems dispensible. The finding that NK cells were less cytolytic and iMOs less efficient in restricting viral dissemination due to IFNAR deficiency in this model is concincingly demonstrated. The expendable role of intrinsic IFNAR signalling for T cell generation/activation is less clear. The manuscript is very well written and the experiments are thoroughly performed. The manuscript contains interesting and novel data. I have however some points that should be addressed.

**Part II – Major Issues: Key Experiments Required for Acceptance**

Reviewer #1: (1) Figure 3G-3K indicate that the absence of IFNAR in the T cells is not detrimental for protective responses using the proportions of reconstituted bone marrow cells indicated (20%). What is the rationale for this proportion? Have these proportions been titrated? Could there be a mixture (let’s say 90% RAG + 10% IFNAR or B6) at which the absence of IFNAR makes the T cell response worse?

(2) It would be informative to measure the actual amounts of IFN in the dLN of IFNAR-delta-Lyz2 animals compared to IFNAR-fl/fl animals. If IFNAR is needed in iMO to enhance IFN-I levels rather than provide antiviral signalling, the defect of IFNAR-delta-Lyz2 animals should be by-passed by exogenous administration of physiological levels of IFN (i.e., resembling those observed in IFNAR-fl/fl animals).

Reviewer #2: (No Response)

Reviewer #3: The authors discuss and expand the relevance of their findings to other viral pathogens and infection models. This however has strong limitations since the footpad infection model is a highly artificial one and does not represent the natural infection route for ECTV. I think the data would be more relevant if the authors could reproduce at least some of their findings in a clinically relevant route for mousepox infection e.g. the intranasal route for respiratory infection. This would exclude route-specific effects and, additionally, support the role of this model as a surrogate model for human smallpox.

**Part III – Minor Issues: Editorial and Data Presentation Modifications**

Reviewer #1: (3) It would be good to emphasise in the discussion that the dispensable role of IFNAR in T cell responses is observed in the context of primary infection and that, although the authors measure CD44+ cells at 22 dpi, results could be different in the context of challenge or others.

(4) A previous paper from the group in PLOS Pathogens showed that cGAS deficiency in parenchymal cells did not affect IFN-I production and resistance to mousepox (being cGAS the main IFN-producing sensor for ECTV). This is now extended here showing that IFNAR is also dispensable in parenchymal cells. Could this poor immune function of hepatocytes explain why the liver is a preferred target organ for ECTV?

(5) In this same previous paper, IFN-I expression in the dLN was unaffected by the absence of IFNAR, which seemingly contradicts the findings presented here. What is the authors’ explanation for these observations?

(6) The IFN-I/IFNAR positive loop is somewhat surprising because IFN itself is not an ISG. However, one could think of indirect mechanisms by which other ISG contribute to enhanced IFN production, like IRF7 or cGAS. The authors should elaborate on this in the Discussion.

(7) The paper is well-written and the data are well-presented, with some minor formatting issues that the production team should easily sort out.

Reviewer #2: (No Response)

Reviewer #3: Could the authors speculate more about the role of IFN I on T cell responses in the ECTV infection model? I think there are known inhibitors of IFN I in ECTV most likely interfering with T cell generation anyway and this must be compensated somehow (e.g. by IL-12 as has been shown for vaccinia VACV?).

In this context, the authors should also restrict the relevance of their data to the acute phase of infection. Memory responses have not been investigated here, although IFN I has a known strong and particular effect on memory cell formation in various other viral infection models.

In this study, T cell responses have been exclusively investigated systemically in spleen and blood. Since the foodpad route is a strictly peripheral one, the T cell responses in the dLNs would be of particular interest, especially with regard to Figure 6 H-J.

Fig. 4H GzmB on x axis is not ledgible.

PLOS authors have the option to publish the peer review history of their article (what does this mean?). If published, this will include your full peer review and any attached files.

Reviewer #1: No

Reviewer #2: No

Reviewer #3: No
---

## [Decision Letter · Decision Letter 1]

28 Apr 2021

Dear Luis,

We are pleased to inform you that your manuscript 'Resistance to lethal ectromelia virus infection requires Type I interferon receptor in natural killer cells and monocytes but not in adaptive immune or parenchymal cells' has been provisionally accepted for publication in PLOS Pathogens.

Best regards,

Gerd Sutter

Guest Editor

PLOS Pathogens

Klaus Früh

Section Editor

PLOS Pathogens

Kasturi Haldar

Editor-in-Chief

PLOS Pathogens

orcid.org/0000-0001-5065-158X

Michael Malim

Editor-in-Chief

PLOS Pathogens

orcid.org/0000-0002-7699-2064

The authors have provided an appropriately revised and clearly improved manuscript. All issues raised by the reviewers have been addressed.

Reviewer Comments (if any, and for reference):

Reviewer's Responses to Questions

**Part I - Summary**

Reviewer #1: (No Response)

Reviewer #3: (No Response)

**Part II – Major Issues: Key Experiments Required for Acceptance**

Reviewer #1: The authors have addressed my major queries and improved the manuscript with further clarifications.

Reviewer #3: (No Response)

**Part III – Minor Issues: Editorial and Data Presentation Modifications**

Reviewer #1: The authors have addressed my minor queries and improved the manuscript with further clarifications.

Reviewer #3: (No Response)

PLOS authors have the option to publish the peer review history of their article (what does this mean?). If published, this will include your full peer review and any attached files.

Reviewer #1: No

Reviewer #3: No

---

## [Editor Report · Acceptance letter]

17 May 2021

Dear Dr. Sigal,

We are delighted to inform you that your manuscript, "Resistance to lethal ectromelia virus infection requires Type I interferon receptor in natural killer cells and monocytes but not in adaptive immune or parenchymal cells," has been formally accepted for publication in PLOS Pathogens.

Best regards,

Kasturi Haldar

Editor-in-Chief

PLOS Pathogens

orcid.org/0000-0001-5065-158X

Michael Malim

Editor-in-Chief

PLOS Pathogens

orcid.org/0000-0002-7699-2064